# RESA: Bringing Back What Sparse Attention Ignores with Residual Estimation

**Weihao Yang**[1]†**, Hao Huang**[1]**, Ningke Li**[1]**, Shihao Wang**[1]**, Darong Yang**[1]**, Yanqi Pan**[1]**,
Wen Xia**[1,2]‡,***Shiyi Li**[1]**, Xiangyu Zou**[1]

[1] Harbin Institute of Technology, Shenzhen, [2] Pengcheng Laboratory
†: weihao.yang00@hotmail.com, ‡: xiawen@hit.edu.cn

## Abstract

Large Language Models (LLM) have gained significant attention. KV cache, stored to avoid quadratic complexity of attention, becomes a bottleneck due to the demands for long-context. Sparse attention (SA) has been proposed to address this by only selecting critical KVs for attention, which may degrade model quality in less sparse scenarios. To improve quality, rather than selecting more KVs, this paper reveals another perspective by estimating the contributions of remaining KVs, called *Residual Estimation*. We find that attention logits (before softmax) exhibit substantial redundancy due to its inherent low-rank nature. We perform Singular Value Decomposition (SVD) on logits matrix in prefilling and find *the spectral dominance of principal singular value* and *its linearly scaling property with sequence length*. These imply that increasing sequence length leads to replication-like logits growth with significant redundancy. However, it is impossible to perform SVD at each decoding step in practice due to its heavy costs. To this end, we propose **RESA**, a training-free framework compensating SA's output with an estimated low-rank prior of logits. RESA introduces (i) a *Prior Estimator* that derives a prior distribution from a typical query as a rank-1 approximation at the end of prefilling, and (ii) an *Online Aggregator* that fuses the prior with SA at each decoding step via lightweight scaling and merging. Besides, we further show that RESA's effect comes from priors being used as attention bias for knowledge injection. Extensive experiments show that without extra overheads, RESA improves model quality by up to 26% across various tasks with the same KV budget compared to state-of-the-arts. Moreover, RESA maintains the same quality with up to 33.2% KV budget reduction and $1.23\times$ attention throughput improvement.

## 1 Introduction

The success of large language models (LLMs) has brought heavy computational overhead due to the quadratic complexity of the attention. KV cache alleviates this by storing intermediate results of $Q, K$ with linear space complexity, thereby avoiding redundant recomputation. However, the growing demand for long-context models leads to a rapid expansion of KVs, becoming a new bottleneck.

Sparse attention (SA) is proposed to mitigate KV cache bottleneck. By exploiting the inherent sparsity in attention, SA selects only the highest-scoring KVs for computation. Existing works mainly focus on two perspectives: (1) *How to precisely identify the critical KVs?* As we cannot globally retrieve attention scores for ideal top-k selection, this leads to the adoption of Approximate Nearest Neighbor (ANN) algorithms to improve selection accuracy. (2) *How many critical KVs should be selected?* As different attention heads exhibit varying sparsity, this leads to dynamic KV budget allocation for each head to avoid excessive resource wastage. However, when facing complex scenarios that attention scores are less sparse, it seems that the only way is to select more KVs to maintain model quality, leading to the gradual loss of SA's advantages, especially on resource-constrained devices. We identify the core problem lies in a common assumption of above perspectives, that is *only selected KVs can contribute to the final attention output, while unselected KVs contribute nothing*.

---

*Corresponding Author Wen Xia.

Figure 1: **Schematic diagram of RESA's effect and comparison between FA, SA and our proposed RESA**. RESA pushes current SA methods to a higher accuracy. It estimates the prior distribution for KVs ignored by SA in a lightweight and accurate way for a more accurate output.

We couldn't help thinking: **even if we don't select more KVs, is there any way we can use the unselected KVs to compensate for the SA output?**

Our work introduces the third, orthogonal perspective: *How to efficiently and accurately estimate the contribution of remaining KVs without selecting them?* We call this process *residual estimation*. Our motivation is simple, that is **there exists significant redundancy in attention calculations, thus the contribution of unselected KVs can be roughly estimated without exact calculation.** Specifically, we reveal the fact of redundancy due to the inherent low-rank nature of the attention logits matrix (before softmax), which is upper bounded by the head dimension and is independent of sequence length. Taking llama3.1-8B as an example, given the sequence length of 8k, the effective rank of attention logits matrix ($\in \mathbb{R}^{8k \times 8k}$) is only 128. Our experiments further show that the principal singular value of attention logits across different settings can account for 40% of the total energy, indicating that few ranks can accurately reconstruct the entire attention logits. This reinforces the rationale of residual estimation for reducing redundant calculations, especially in long-context cases.

To this end, we propose **RESA** (**R**esidual **E**stimated for **S**parse **A**ttention), a training-free framework to compensate SA. As shown in Figure 1, RESA can push existing SA methods closer to a more ideal attention mechanism with higher accuracy and less budget. RESA captures low-rank structure during prefilling and estimates unselected KVs' logits (i.e., prior distribution) for sparse decoding. RESA has two modules: the *Prior Estimator* to determine the logits of remaining KVs, and the *Online Aggregator* to compensate for sparse attention outputs. To avoid the heavy overhead of online SVD, the *Prior Estimator* finds a representative query to approximate the contribution of principal singular value and precomputes with all keys as a prior during prefilling. The *Online Aggregator* then integrates the estimated residuals with the sparse attention output in a lightweight manner. By rescaling and merging the results in a delta-manner, the computational complexity maintains the same as vanilla SA. These two modules ensure RESA to improve the accuracy of SA without introducing additional overhead. Finally, we explore the reasons for RESA's effectiveness and find that the prior distribution injects knowledge from the remaining KVs by imposing biases to the attention. We further analyze how it can guide possible model optimization at training-time.

In summary, our contributions are as follows:

- We identify the redundancy of attention logits due to its inherent low-rank nature, providing a new perspective to estimate residual ignored by sparse attention for compensation.

- We propose RESA, a training-free framework capturing low-rank structure in prefilling and estimate residual for decoding. RESA uses the *Prior Estimator* to estimate a prior distribution for residual, and the *Online Aggregator* to merge it with sparse attention lightly.

- We further reveal that RESA's effect comes from the prior being used as attention biases for knowledge injection, which is worth exploring more of its features at training-time.

We conducted various experiments and verify RESA's effects. Given RULER and LongBench benchmark, RESA can improve the accuracy of a single task by up to 26%. Such improvements can be scaled to different context length. RESA also reduces the error of attention scores between full attention and sparse attention by up to 77%. When maintaining the same model quality, RESA can reduce the KV budget by up to 33.2% and improve attention throughput by 1.23$\times$.

## 2 RELATED WORKS

**Long-context Model.** Long-context LLMs have become popular to support complex applications like CoT Wei et al. (2022), TOT Yao et al. (2023) and GoT Besta et al. (2024). Therefore, existing LLMs usually use Rotary Position Embeddings (RoPE) Su et al. (2024) and YaRN Peng et al. (2024) to extend their context windows beyond the initial training limits. For example, OpenAI GPT-4.1 OpenAI (2023) supports a context window of up to 1 million tokens, Claude 3 cla offers a context window of 200K, Gemini 1.5 Pro Team et al. (2024) features a context length in the millions, and Kimi-VL Team et al. (2025) demonstrates strong performance with a 128K context window.

**Efficient Attention and Sparse Attention.** vLLM Kwon et al. (2023) uses page to efficiently manage the KV cache. FlashAttention series Dao et al. (2022); Dao (2024); Shah et al. (2024) co-design with hardware to minimize I/O cost for acceleration. Some other methods He & Zhai (2024); Hong et al. (2024); Jiang et al. (2024); Aminabadi et al. (2022) also overlaps computation and I/O for efficient inference. For KV selection in sparse attention, StreamingLLM Xiao et al. (2024b) handles long context with attention sinks. TOVA Oren et al. (2024) permanently discard tokens based on the current query. The statical decision on important KV cache leads to performance degradation. H2O Zhang et al. (2023) and FastGen Ge et al. (2024) uses the historical attention scores and sophisticated strategy to identify the important KV. SparQ Ribar et al. (2024), infiniGen Lee et al. (2024), infLLM Xiao et al. (2024a) and IMPRESS Chen et al. (2025a) maintain all KV and calculate approximate attention scores for selection. PSA Zhou et al. (2025) and Twilight Lin et al. (2025) set the threshold of accumulated attention scores to dynamically decide KV budget. Other methods like magicpig Chen et al. (2025b) performs token-level LSH-based sampling to select KV.

## 3 BACKGROUND AND MOTIVATION

### 3.1 SPARSE ATTENTION WITH KV CACHE

Given matrices: $\mathbf{Q} \in \mathbb{R}^{l_q \times d}, \mathbf{K} \in \mathbb{R}^{l_{kv} \times d}$ and $\mathbf{V} \in \mathbb{R}^{l_{kv} \times d}$, where $l_q, l_{kv}, d$ denotes the sequence length of query, key/value, and head dimension respectively. The attention can be written as

$$\mathbf{M} = \mathbf{Q} \cdot \mathbf{K}^\top / \sqrt{d}, \quad \mathbf{S} = \text{softmax}\,(\mathbf{M}) \in \mathbb{R}^{l_q \times l_{kv}}, \quad \mathbf{O}_{FA} = (\mathbf{S} \cdot \mathbf{V}) \in \mathbb{R}^{l_q \times d}, \quad (1)$$

where $\mathbf{M}, \mathbf{S}$ are called attention logit and score, respectively. $\mathbf{S}_{i,j}$ reflects the importance between $i$-th and $j$-th token. The process of LLM inference is split into **prefilling** and **decoding** stages, where prefilling takes user requests as tokens to generate the first token, and decoding takes the latest token to generate new tokens step-by-step. Note that during the decoding stage, the $\mathbf{K}, \mathbf{V}$ will be appended by newly generated key value states. Due to the huge computational cost, **S**parse **A**ttention (SA) selects only critical KVs according to given budgets. Specifically, a common SA method partitions all KVs into blocks and selects KVs at the block level. Each block is represented by compact metadata (e.g., bounding cuboid estimation). Therefore, block-level relevance is approximated by dot products between metadata and queries. The top-K relevant blocks are retrieved for SA computation. Our paper focuses on the SA methods applying in decoding stage.

### 3.2 MOTIVATION ON RESIDUAL ESTIMATION

The residual used to correct SA requires a prior logit distribution assigned to the positions that are not selected by SA. Fortunately, the prior distribution can be approximated without obtaining actual $K$s, which is supported by both theoretical analysis and experimental observations.

**Analysis 1: Low-Rank Basis of Attention Logits Matrix.** Previous studies Singhania et al. (2024); Bhojanapalli et al. (2020); Sanyal et al. (2024) suggest that the attention logit matrix $M = Q \cdot K^T / \sqrt{d}, M \in \mathbb{R}^{l_q \times l_{kv}}$ exhibits an inherent low-rank structure. Specifically, the minimal dimension of $Q, K$ is the head dimension $d$, thus the effective rank of $M$ is upper bounded by $d$ (e.g., 128), which is much smaller than the sequence length $l_q$ or $l_{kv}$ (e.g., 8k). This strongly indicates that as the sequence length increases, the full attention map still lies in a much smaller subspace, implying substantial redundancy in the original attention computation.

**Observation 1: Dominance of the Principal Singular Value in Attention Logits Matrix.** To further observe the low-rank structure, we conducted singular value decomposition (SVD) of attention logits matrices across different layers, heads and sequence lengths, which can be formulated as

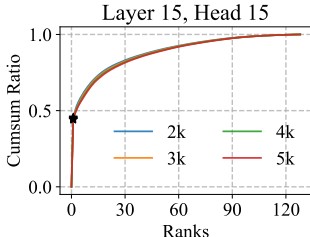 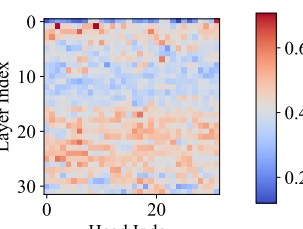 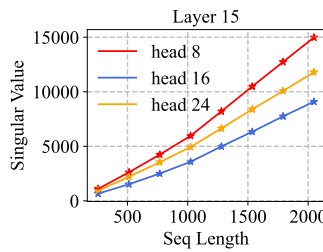

(a) Accumulate ratio of singular values with various context length.

(b) The ratio of principal singular value across layers and heads.

(c) Linear scaling of singular values with different context length.

Figure 2: **Llama-3.1-8B's redundancy in attention calculations from low rank perspective.** (a) shows the energy contribution of singular values, whose sparsity remains nearly constant across different context lengths. (b) further shows the percentage of the total energy of the main singular values across layers and heads, which remains stable at 40%-50%. (c) shows that the size of the main singular values increases almost linearly with context length.

$M = U\Sigma V^T$. We first observed the obvious dominance of the principal singular value, as shown in Figure 2(a). We then observed this phenomenon across layers and heads, and a consistent pattern emerges: the principal singular value alone almost accounts for *40%-50% of the total sum of singular values*, as shown in Figure 2(b). Intuitively, this implies that a coarse approximation of attention logits matrices can be reconstructed with very few ranks, and the first rank contributes almost half of the basic global pattern between different queries and keys.

**Observation 2: Linear Scaling of the Principal Singular Value with Sequence Length.** As illustrated in Figure 2(c), we find that the principal singular value grows approximately linearly with the sequence length $l_{kv}$. Intuitively, this scaling behavior indicates that as more tokens are introduced, the dominant global structure simply amplifies without fundamentally altering its orientation. In other words, with the expansion of sequence length, the logit matrix expands in a replication-like manner. For example, given matrix $M_1 \in \mathbb{R}^{L \times L}$ with sequence length $L$, when sequence length grows to $2L$, the corresponding logit matrix $M_2 \in \mathbb{R}^{2L \times 2L}$ can be approximated as $M_2 \approx \begin{bmatrix} M_1 & M_1 \\ M_1 & M_1 \end{bmatrix}$, which implies the main structure of attention logits is persistent and predictable.

**Beyond Sparsity: A Low-Rank Perspective for Compensating Sparse Attention.** The above analysis and observations reveal the significant low-rank nature of attention logits and its replication-like expansion pattern. Therefore, attention inherently has a lot of redundant calculations and can be approximated by low-rank structures. However, most current efficient attention methods primarily rely on sparsity to approximate precise attention logits (i.e., sparse attention), selecting top-k logits and treating remaining KVs as zero contributions. Indeed, sparsity can capture salient fine-grained features more accurately, but it ignores the global low-rank structure of the attention logits. The core significance of our work lies in fully combining these two different perspectives. They are not mutually exclusive but complementary: sparse approximations exploit fine-grained importance, whereas low-rank approximations preserve global structure of attention. Together, they provide an opportunity to approximate the full attention logits more accurately.

**Main Challenges.** However, directly extracting singular values and vectors via online SVD is computationally prohibitive during inference. Thus the key challenge is to design lightweight and accurate estimators of the dominant rank structure without explicit decomposition. Addressing this challenge is central to bridging the gap between theoretical low-rank properties and practical estimation methods, which leads to our method – **R**esidual **E**stimation for **S**parse **A**ttention (RESA).

## 4 METHODOLOGY

### 4.1 RESA OVERVIEW

**General Framework of RESA.** As shown in Figure 3(a), RESA adds a new computational branch to the vanilla SA process. RESA mainly consists of two submodules, the *Prior Estimator* determines a *prior distribution* of attention logits and calculates its corresponding $O_{Est}$, and the *Online Aggregator* combine the priors and SA's result with scaling factors to get a more accurate output.

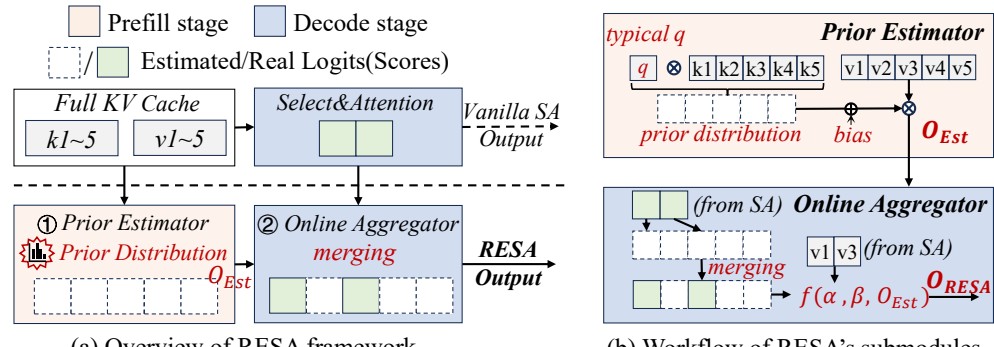

(a) Overview of RESA framework.  (b) Workflow of RESA's submodules.

Figure 3: **RESA framework and the workflow of its sub-modules.** (a) shows how RESA compensates SA with two sub-modules: The *Prior Estimator* lightly estimate the prior distribution, and then the *Online Aggregator* merges it with the result of SA. (b) shows the details of two sub-modules. The *Prior Estimator* uses a typical $q$ to estimate prior distribution at the end of prefilling. The *Online Aggregator* uses two scaling factors $\alpha, \beta$ and reuses results of Estimator for lightweight merging.

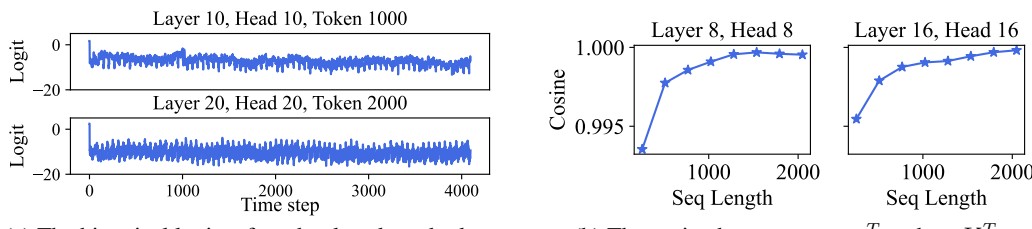

(a) The historical logits of randomly selected tokens.  (b) The cosine between $\sigma_1 u_1 v_1^T$ and $\mu_Q K^T$.

Figure 4: **The phenomenon that historical logits are concentrated around the mean and the cosine between $\sigma_1 u_1 v_1^T$ and $\mu_Q K^T$.** This indicates that $\mu_Q K^T$ can be used as a very lightweight and accurate approximation of the first rank's contribution.

Note that RESA does not focus on specific sparse attention algorithms or system optimizations. As a general framework, RESA mainly contributes to how to enhance different sparse attention methods in a lightweight but effective way.

## 4.2 PRIOR ESTIMATOR: RANK-1 PRIOR LOGITS APPROXIMATION

**Goal.** Due to the heavy cost of SVD, we can not apply it at each decoding step, so it is difficult to directly estimate the contribution of different ranks. Considering this, we simplify the problem and only estimate the contribution of the principal singular value, which can already provide a meaningful prior. Thus, our goal is to (i) accurately approximate the contribution of principal SVD component, and (ii) efficiently calculate the corresponding output $\mathbf{O}_{Est}$ of the prior.

**Approximating the Contribution of Principal SVD component.** Our estimation method is

$$\sigma_1 u_{1,i} v_1^T \approx \mu_Q K^T, q_i K^T \approx \mu_Q K^T + (q_i - \mu_Q)\mu_K^T, \delta_i = (q_i - \mu_Q)(K - \mu_K)^T \quad (2)$$

where $q_i, K$ denote the $i$-th query and all Key caches ($\forall k_j$) at current decoding step, $\mu_Q, \mu_K$ denote the mean of previous queries and keys, $\sigma_1 u_{1,i} v_1^T$ denotes the first rank's contribution of $q_i$, which is called the principal SVD component. $\delta_i$ denotes the error between true and estimated logits. Our estimation consists of a *prior term* (approximation to rank-1) and a *bias term*. **For the prior term $\mu_\mathbf{Q} \mathbf{K}^\mathbf{T}$, it represents the mean of historical logits, which aligns closely with the principal SVD component $\sigma_1 \mathbf{u_{1,i}} \mathbf{v_1^T}$.** Given the true logits matrix $M$, the $i$-th row of it is $M_{i,:} = q_i K^T$, our prior $\mu_Q K^T = (\frac{1}{L} \sum_{i=1}^{L} q_i) K^T = \frac{1}{L} \sum_{i=1}^{L} M_{i,:} = \overline{M}$, representing the mean value across the entire prefilling attention logits for all Ks. Using this as the prior is because that we observed the historical logits of each token (i.e., a column of logits matrix) exhibit a concentrated structure: these logits mostly revolve around their mean logit with relative small fluctuations, as shown in Figure 4(a). Thus, we can represent the $i$-th row of logits matrix as $M_{i,:} = \overline{M} + e_i$, where $e_i$ is the small fluctuations of $i$-th row. This leads to the *rank-1 spike* phenomenon in attention logits matrix, thus

the mean of rows $\overline{M}$ (i.e., our prior $\mu_Q K^T$) is almost aligning with the principal SVD component. In Figure 4(b), we show the cosine similarity between $\sigma_1 u_{1,i} v_1^T$ and $\mu_Q K^T$, which is very close to 1 and proves the effectiveness. We further use **a bias term $(\mathbf{q_i} - \mu_\mathbf{Q}) \cdot \mu_\mathbf{K}^\mathbf{T}$ to align the mean of true logits** $q_i \cdot K^T$ **and the prior, avoiding the distortions for the following Aggregation.** Considering softmax is very sensitive to the relative magnitude of logits, this bias eliminates global shifts when aggregating with logits selected by SA, which is crucial to avoid systematic distortions of final output. More detailed discussions can be found in Appendix.

**Efficient Computation of Prior.** An important property of our estimation is that the computation of prior distribution $\mu_Q \cdot K^T$ is independently of the query index $i$, indicating it can be pre-computed once, rather than recomputed at each decoding step. As shown in Figure 3(b), specifically, at the end of prefilling, we compute the prior distribution, apply softmax to it, and directly use it to weight all $V$s as output $\mathbf{O}_{Est}$. This is equivalent to appending an extra query at the end of the prefilling, that is changing the context length from $L$ to $L+1$. Since prefilling typically already has very large $L$, the increased overhead is negligible compared to the overall cost, yet provides the necessary prior to *all subsequent decoding steps.* Note that in practice, we use the chunked prefill Agrawal et al. (2023) to avoid OOM and merge the calculation of prior into the last chunk to fully utilize computing power, indicating that only the size of the last chunk is increased by 1.

## 4.3 Online Aggregator: Rescaling and Delta Merging

**Naive Aggregation.** For each decoding step, the Aggregator aims to merge the priors and SA for a more accurate output. An intuitive way is to use the logits obtained by SA to replace the logits at the corresponding position of priors. Given prior logits and the corresponding scores $\mathbf{P}, \mathbf{S} \in \mathbb{R}^{B \times H \times L}$, where $B, H, L$ denotes batch size, the number of heads and the sequence length, respectively. All real logits computed by SA is denoted as $\mathbf{P}_\mathcal{I}^{SA} \in \mathbb{R}^{B \times H \times |\mathcal{I}|}$ and their indices construct a set $\mathcal{I} \subseteq \{1, \ldots, L\}$. Thus, the aggregated logits $\mathbf{P}'$ and the corresponding output $\mathbf{O}_{RESA}$ can be written as

$$\mathbf{P}'_j = \begin{cases} \mathbf{P}_{\mathcal{I}j}^{SA}, & j \in \mathcal{I}, \\ \mathbf{P}_j, & j \notin \mathcal{I}, \end{cases} \quad \mathbf{S}' = \text{softmax}(\mathbf{P}'), \quad \mathbf{O}_{RESA} = (\mathbf{S}' \cdot \mathbf{V}). \quad (3)$$

This naive way requires $O(L)$ complexity, which is the same as FA and loses advantages of SA. Thus, we need to derive an efficient aggregation with $O(|\mathcal{I}|)$ complexity, which is the same as SA.

**Online Aggregation: Rescale.** Let $Z = \sum_j \exp(\mathbf{P}_j)$ be the exponential sum of prior, and $Z' = \sum_j \exp(\mathbf{P}'_j)$ be the new one. $Z$ can be pre-calculated only once in the prefilling stage with the prior calculation, avoiding online heavy costs. For $Z'$, the only difference between $Z$ is to replace $\sum_{j \in \mathcal{I}} \exp(\mathbf{P}_j)$ with $\sum_{j \in \mathcal{I}} \exp(\mathbf{P}_{\mathcal{I}j}^{SA})$, which only requires $O(|\mathcal{I}|)$ complexity (like SA) without any matrix multiplications. Thus $Z'$ can be obtained online lightly. Given $Z$ and $Z'$, the rescale step uses two scaling factors to align the prior and SA's scores with $S'$, which can be written as

$$\mathbf{S}'_j = \begin{cases} \alpha \cdot \mathbf{S}_j, j \notin \mathcal{I}, \\ \beta \cdot \mathbf{S}_{\mathcal{I}j}^{SA}, j \in \mathcal{I}, \end{cases} \quad \mathbf{S}_\mathcal{I}^{SA} = \text{softmax}(\mathbf{P}_\mathcal{I}^{SA}), \quad \alpha = \frac{Z}{Z'}, \quad \beta = \frac{\sum_{j \in \mathcal{I}} \exp(\mathbf{P}_{\mathcal{I}j}^{SA})}{Z'}. \quad (4)$$

**Online Aggregation: Delta Merging.** According to Eq 4, we can rewrite the $\mathbf{O}_{RESA}$ in Eq 3 as $\mathbf{O}_{RESA} = \sum_{j \notin \mathcal{I}} \alpha \cdot \mathbf{S}_j \cdot \mathbf{V}_j + \sum_{j \in \mathcal{I}} \beta \cdot S_{\mathcal{I}j}^{SA} \cdot \mathbf{V}_j$. To avoid online $O(L)$ complexity and reuse the result of Estimator $\mathbf{O}_{Est}$, $\mathbf{O}_{RESA}$ can be written in a delta-update manner, which is expressed as

$$\mathbf{O}_{RESA} = \sum_{\forall j} \alpha \mathbf{S}_j \mathbf{V}_j - \sum_{j \in \mathcal{I}} \alpha \mathbf{S}_j \mathbf{V}_j + \sum_{j \in \mathcal{I}} \beta \mathbf{S}_{\mathcal{I}j}^{SA} \mathbf{V}_j = \alpha \cdot \mathbf{O}_{Est} + \left( \beta \cdot \mathbf{S}_\mathcal{I}^{SA} - \alpha \cdot \mathbf{S}_\mathcal{I} \right) \mathbf{V}_\mathcal{I}, \quad (5)$$

where $\mathbf{V}_\mathcal{I}$ is the $V$ caches corresponding to the indices in the set $\mathcal{I}$. Eq 5 maintains the same complexity $O(|\mathcal{I}|)$ as SA. The above process can is summarized in Figure 3(b). Note that for numerical stability, we use the log-sum-exp tricks and the shifted logit tricks (minus the largest logit) to avoid overflow, which are widely used in previous works Dao et al. (2022); Dao (2024); Shah et al. (2024).

**Hyper-parameter for Controlling the Importance of Priors.** To cope with the changing task scenarios, we further introduce a hyperparameter $\lambda \in [0, 1]$ to control the importance of the prior. Specifically, we use $\alpha' = \lambda \cdot \alpha$ to replace the $\alpha$ in Eq 5. When $\lambda$ equals to 0, RESA degenerates to the vanilla SA. More details can be found in our experiments §5.4.

Table 1: Model quality verification of RULER benchmark on Llama-3.1-8B-128k, Llama-3.2-3B-128k, Mistral-7B-512k and LWM-Text-7B-128K models. For each SA method, we compare it with its RESA-enhanced version (+RESA). RESA can improve accuracy by up to 26% for a single task.

| Methods | Model | Niah2 | Niah3 | MK2 | MK3 | MV | MQ | QA |
|---|---|---|---|---|---|---|---|---|
| Quest(**+RESA**) | **Llama-3.1-8B-128k** | 100(**100**) | 99(**100**) | 87(**91**) | 24(**40**) | 89.5(**96**) | 97(**98**) | 86(**87**) |
| ArkVale(**+RESA**) | **Llama-3.1-8B-128k** | 100(**100**) | 100(**100**) | 90(**93**) | 27(**43**) | 88(**94**) | 97.25(**99**) | 86(**87**) |
| Ideal-TopK(**+RESA**) | **Llama-3.1-8B-128k** | 100(**100**) | 100(**100**) | 100(**100**) | 98(**100**) | 98.5(**99.75**) | 99.25(**99.75**) | 86(**87**) |
| Quest(**+RESA**) | **Llama-3.2-3B-128k** | 91(**92**) | 58(**84**) | 74(**82**) | 3(**6**) | 71.75(**84.25**) | 83.25(**93**) | 59(**62**) |
| ArkVale(**+RESA**) | **Llama-3.2-3B-128k** | 89(**89**) | 84(**97**) | 78(**79**) | 3(**6**) | 77(**88**) | 90.5(**94**) | 62(**62**) |
| Ideal-TopK(**+RESA**) | **Llama-3.2-3B-128k** | 95(**95**) | 100(**100**) | 99(**99**) | 71(**92**) | 92.75(**97**) | 97.75(**98.5**) | 63(**63**) |
| Quest(**+RESA**) | **Mistral-7B-512k** | 99(**99**) | 71(**93**) | 92(**93**) | 9(**27**) | 94(**94.5**) | 71(**74**) | 83(**87**) |
| ArkVale(**+RESA**) | **Mistral-7B-512k** | 100(**100**) | 95(**99**) | 96(**96**) | 28(**44**) | 80.5(**80.75**) | 97.5(**98**) | 87(**92**) |
| Ideal-TopK(**+RESA**) | **Mistral-7B-512k** | 100(**100**) | 100(**100**) | 100(**100**) | 99(**99**) | 93.75(**94**) | 100(**100**) | 90(**90**) |
| Quest(**+RESA**) | **LWM-Text-7B-128k** | 73(**74**) | 89(**90**) | 25(**26**) | 2(**3**) | 75.5(**82**) | 76(**84**) | 28(**30**) |
| ArkVale(**+RESA**) | **LWM-Text-7B-128k** | 75(**75**) | 90(**90**) | 29(**30**) | 2(**3**) | 80(**83.25**) | 74.75(**89.25**) | 30(**30**) |
| Ideal-TopK(**+RESA**) | **LWM-Text-7B-128k** | 98(**98**) | 91(**91**) | 93(**94**) | 50(**70**) | 83(**83**) | 88.5(**92.5**) | 41(**41**) |

## 4.4 PRIOR DISTRIBUTION IMPOSES ATTENTION BIAS FOR KNOWLEDGE INJECTION

The core of RESA is to determine the prior distribution, which can be viewed as a bias in the attention mechanism. This bias manifests in two ways: (1) The prior distribution directly influences the attention scores. Specifically, in the softmax calculation, the prior distribution becomes part of the denominator. This adjustment allows the prior distribution to shape the distribution of attention scores. (2) The weighted sum of V caches determined by the prior distribution will be aggregated to the final attention output (like RESA's compensation for sparse attention). Thus, the prior distribution serves to adjust both the attention scores and the final output.

**Diversity of Prior's Effects.** Beyond RESA's training-free compensation for SA, the prior distribution can be also leveraged for additional purposes at training time. *Case 1*: OpenAI's recent open-source model GPT-oss introduces a learnable bias for sparse attention layers, which is discarded after the softmax computation. This can be interpreted as the model is learning an implicit prior distribution during training, effectively marking certain positions as irrelevant by setting their output to zero, thus guiding the model to focus on more meaningful positions and improving the model quality. *Case 2*: Previous work Sun et al. (2024) introduces a learnable $K$ and $V$ to reduce extreme outliers in LLM activations, which can be explained as learning an implicit prior that evenly spreads the impact of large outliers across all positions, mitigating their disproportionate influence.

## 5 EVALUATION

### 5.1 EXPERIMENTAL SETUP

**Testbed.** Our experiments were conducted on a host equipped with a single 3090 GPU with 24GB HBM, an Intel(R) Xeon(R) Gold 5218 CPU @ 2.30GHz, and 128GB of DRAM. The GPU and CPU are connected via PCIe 4, which provides the bandwidth of 32GB/s.

**Models.** Our evaluated models include Llama-3.1-8B Grattafiori et al. (2024) with 128k context length, Llama-3.2-3B Grattafiori et al. (2024) with 128k context length, Mistral-7B Jiang et al. (2023) with 512k context length and LWM-Text-7B Liu et al. (2025) with 128k context length. We use two Llama models with different sizes to show the generalization across model size, then use Mistral and LWM model to show the generalization across model types. These models include both Multi-Head Attention (MHA) and Group-Query Attention (GQA).

**Benchmark.** Our evaluations are tested on the LongBench Bai et al. (2024) and RULER Hsieh et al. (2024). LongBench is a famous benchmark built on diverse real-world long-context tasks, covering 6 task categories like document QA, summarization, and few-shot learning. Its primary goal is to evaluate how well models can utilize extended contexts in practical applications. RULER is another famous benchmark that generates synthetic examples of 13 tasks to evaluate long-context language models. RULER can configure task complexity in a friendly and flexible way, thus more settings can be configured to evaluate the effect of RESA. Together, they provide complementary perspectives: LongBench tests utility in practice, whereas RULER examines capability in controlled settings.

Table 2: Model quality verification of Longbench benchmark on Llama-3.1-8B-128k, Llama-3.2-3B-128k, Mistral-7B-512k and LWM-Text-7B-128K. For each SA method, we compare it with its RESA-enhanced version (+RESA). RESA improves accuracy by up to 8.22% for a single task.

| Methods | Model | 2WMQA | Qas | MFQA | GR | MN | PR |
|---|---|---|---|---|---|---|---|
| Quest(**+RESA**) | **Llama-3.1-8B-128k** | 5.26(**6.36**) | 10.88(**10.98**) | 22.43(**22.58**) | 24.55(**24.95**) | 21.68(**21.68**) | 98.39(**98.39**) |
| ArkVale(**+RESA**) | **Llama-3.1-8B-128k** | 5.31(**6.52**) | 10.25(**10.36**) | 21.65(**22.06**) | 24.59(**24.65**) | 21(**21.68**) | 89.19(**94.84**) |
| Ideal-TopK(**+RESA**) | **Llama-3.1-8B-128k** | 5.23(**6**) | 10.59(**10.99**) | 20.89(**23.23**) | 24.91(**25.11**) | 21.1(**21.1**) | 98.39(**98.39**) |
| Quest(**+RESA**) | **Llama-3.2-3B-128k** | 15.82(**17.77**) | 34.01(**34.28**) | 49.44(**49.78**) | 23.7(**23.91**) | 20.53(**20.71**) | 90.86(**90.97**) |
| ArkVale(**+RESA**) | **Llama-3.2-3B-128k** | 15.96(**16.23**) | 36.17(**36.57**) | 49.49(**51.33**) | 22.75(**23.23**) | 20.8(**21.16**) | 85.93(**87.63**) |
| Ideal-TopK(**+RESA**) | **Llama-3.2-3B-128k** | 20.34(**20.64**) | 35.34(**36.32**) | 50.62(**50.8**) | 23.37(**24.63**) | 20.82(**20.93**) | 91.4(**91.4**) |
| Quest(**+RESA**) | **Mistral-7B-512k** | 22.41(**30.63**) | 10.6(**10.79**) | 32.48(**38.56**) | 19.97(**20.07**) | 19.56(**19.76**) | 100(**100**) |
| ArkVale(**+RESA**) | **Mistral-7B-512k** | 24.9(**27.18**) | 10.88(**11.1**) | 33.89(**38.54**) | 20.43(**20.64**) | 19.6(**19.67**) | 100(**100**) |
| Ideal-TopK(**+RESA**) | **Mistral-7B-512k** | 27.13(**30.3**) | 11.27(**11.67**) | 33.28(**36.91**) | 20.06(**20.17**) | 20.67(**20.67**) | 100(**100**) |
| Quest(**+RESA**) | **LWM-Text-7B-128k** | 3.67(**4.08**) | 7.91(**8.89**) | 12.86(**12.89**) | 18.63(**18.87**) | 10.03(**12.55**) | 3.76(**9.77**) |
| ArkVale(**+RESA**) | **LWM-Text-7B-128k** | 3.79(**4.23**) | 7.14(**8.65**) | 13.19(**13.54**) | 19.34(**19.42**) | 10.2(**12.38**) | 3.76(**8.26**) |
| Ideal-TopK(**+RESA**) | **LWM-Text-7B-128k** | 4.06(**4.48**) | 8.24(**10.05**) | 13.95(**14.82**) | 18.49(**19.66**) | 12.44(**12.8**) | 10.6(**12.68**) |

**Compared Methods.** We first compare RESA with Quest Tang et al. (2024), ArkVale Chen et al. (2024), and Ideal Top-K. Quest and ArkVale are popular SA methods that take into account both system efficiency and accuracy in identifying critical KV pages. Ideal Top-K is a token-grained SA method, which identifies the most important $k$ KVs and can be considered as the theoretical upper bound. We then show how KV budgets are reduced with RESA and adopt to Ada-KV Feng et al. (2024) and PSA Zhou et al. (2025), which automatically determine KV budget for each head.

**Additional Configuration.** Due to the limited GPU memory, we use chunked prefill to avoid out-of-memory (OOM) error. Specifically, the chunk size is 1024. When given a 16K length input, it will be split into 16 chunks and perform prefill 16 times. For the KV cache layout, we follow existing works Li et al. (2024); Chen et al. (2025b); Tang et al. (2024); Xiao et al. (2024a) to use the first 4 tokens (initial token) and the last 64 tokens (local token) as a fixed budget. Note that unlike previous works Chen et al. (2024); Tang et al. (2024); Chen et al. (2025b) preserve certain layers as full attention, we use sparse attention for all layers.

## 5.2 ACCURACY IMPROVEMENT

**Evaluation.** We conducted various experiments on LongBench and RULER to show generalization capability of RESA. Results are shown in Table 1 and Table 2. We only show the result of the budget 2.5% under 8K context. More results can be found in the appendix. RESA improves accuracy of a single task by up to 16%, 26%, 22%, 20% for four models on RULER and 5.65%, 1.95%, 8.22%, 6.01% on LongBench, respectively. Ideal-Topk has the smallest improvement because it is already very close to the full attention. We further show RESA's improvement with different context length in Figure 5(a). Generally speaking, the longer the context, the larger the improvement brought by RESA. This is mainly because the longer the context, the more redundant calculations are in the attention, which can be effectively captured by RESA. Note that tasks in LongBench are real-world tasks whose length cannot be controlled. Thus we mainly use RULER for more analysis.

**A Deeper Relationship between RESA's Low-rank Prior and RoPE.** We further delve into understanding the effectiveness of RESA. We visualized the typical query $\mu_Q$ and keys $\mu_K$, as shown in Figure 5(b). Empirically, the high-frequency channels are almost zero because they are canceled out when averaged across sequence, leaving only the low-frequency channels intact (like a low-pass filter), which indicates that RESA's prior distribution is almost calculated by low-frequency signals. Surprisingly, this is consistent with the findings of previous work Barbero et al. (2025), arguing that low-frequency components in RoPE primarily serve as semantic carriers. Thus, *RESA's prior can be considered as a global semantic bias that guides the estimation of logits for unselected KVs.* For more details about the selection of typical query $\mu_Q$, please refer to the Appendix.

## 5.3 ERROR ANALYSIS

**The Error of Attention Scores between FA and SA (w/ and w/o RESA).** Take Quest as an example, Figure 6(a) shows the averaged error of all layers. For almost all heads, compared to vanilla SA, RESA enhanced version can reduce the error about 55%-70%. This is because the prior provides contributions of unselected KVs, which is more accurate than directly setting them to zero.

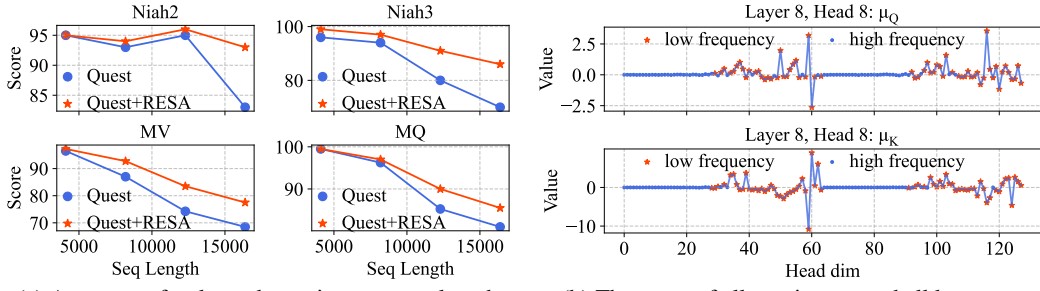

(a) Accuracy of tasks under various context length.

(b) The mean of all queries $\mu_Q$ and all keys $\mu_K$.

Figure 5: **Improvement of RESA with various context length and the visualization of $\mu_Q$ and $\mu_K$.** (a) shows that the longer the context, the greater the improvement brought by RESA. (b) shows the specific value of $\mu_Q$ and $\mu_K$. Their high-frequency channels are almost cancelled out, which indicates RESA's low-rank prior mainly constructed by low-frequency signals in RoPE.

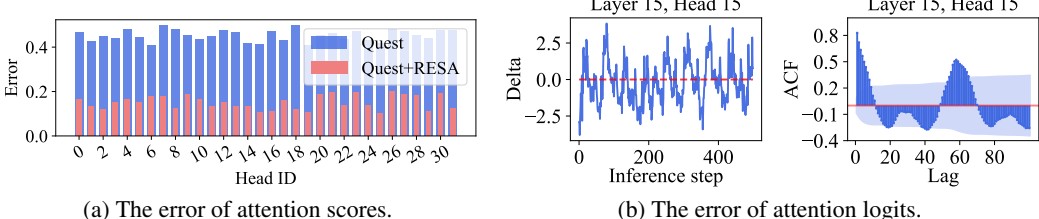

(a) The error of attention scores.

(b) The error of attention logits.

Figure 6: **Error Analysis of attention scores and logits.** (a) shows the layers' average error between FA's attention scores and SA's scores. RESA reduce the error significantly. (b) shows the error between real and estimated logits. The left one shows obvious mean reversion property. The right one plots the autocorrelation function (ACF) of it, revealing temporal dependencies or periodicity.

**The Error between RESA's Estimated Logits and Real Logits.** According to the Eq 2, the error can be written as $\mathbf{E} = (q_i - \mu_Q) \cdot (K - \mu_K)^T$. Figure 6(b) visualizes the error at each inference step, we find an obvious mean reversion property. Thus we consider it as time series data and further examine its temporal dependency with autocorrelation function (ACF). We plot the ACF values over different lags, where the blue bars indicate the correlation coefficients and the shaded region marks the 95% confidence interval under the null hypothesis of white noise. A significant spike at a given lag suggests strong linear dependence between the series and its lagged version. This suggests that the sequence is not independently distributed, but instead retains substantial temporal structure that worth future exploring. More details can refer to the appendix.

## 5.4 HYPER-PARAMETER TUNING

We find two patterns how hyper-parameter $\lambda$ affects accuracy. Firstly, for most retrieval tasks, the accuracy increases as $\lambda$ increases, as shown in Figure 7(a) left. We guess that in this type of task, the prior may help eliminate incorrect candidates, thereby improving model's confidence of retrieval results. For the second case, mainly in summary or comprehension tasks, the accuracy first increases slightly and then drops rapidly as $\lambda$ increases, as shown in Figure 7(a) right. We guess that in this type of task, the prior may interfere model's understanding of the context, so it can only answer through intrinsic parameter knowledge. A typical example is the CWE task in RULER, where the model outputs the most common words of training corpus rather than context.

## 5.5 EFFICIENCY ANALYSIS

**KV Budget Reduction and Throughput Improvements.** We limit the accuracy loss within 2% and see how RESA can reduce the KV budget and improve throughput. We use PSA and Ada-KV to determine the KV budget, results are shown in Figure 7(b) left side. The first three clusters of bars represent PSA, while the remaining represent Ada-KV. For a single task, RESA can further reduce the KV budget by up to 33.2% and 28.7%, respectively. Figure 7(b) right side shows the attention throughput of corresponding tasks, which achieve up to $1.23\times$ and $1.16\times$ improvement compared to PSA and Ada-KV, and $2.64\times$ and $2.49\times$ improvement compared to vanilla attention.

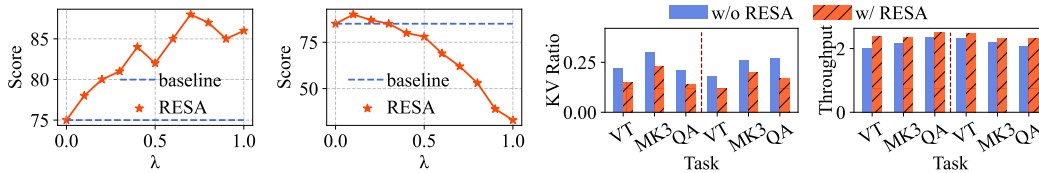

(a) Two patterns of how $\lambda$ affects accuracy.   (b) Efficiency improvement of PSA and Ada-KV.

Figure 7: **Hyper-parameter tuning and efficiency improvement.** (a) shows that the importance of the prior (controlled by $\lambda$) can be beneficial or detrimental in different tasks. (b) shows that RESA can achieve a smaller budget and higher attention throughput.

Table 3: The layer-wise prefilling latency (ms) of RESA using Llama-3.1-8B under varying context length. We demonstrate in two ways: the impact on the last chunk and the entire prefilling process.

| Measured ways | The last chunk of prefilling | | | | | The entire process of prefilling | | | | |
|---|---|---|---|---|---|---|---|---|---|---|
| context length | 8k | 16k | 32k | 64k | 128k | 8k | 16k | 32k | 64k | 128k |
| w/o RESA | 6.8 | 13.3 | 25 | 48.7 | 96 | 27.3 | 106.3 | 399.8 | 1554.8 | 6140.4 |
| w/ RESA | 7.3 | 14 | 26.2 | 51.1 | 100.6 | 27.8 | 106.9 | 400.8 | 1556.7 | 6143.8 |
| extra overhead ratio | 7.35% | 5.26% | 4.80% | 4.93% | 4.79% | 1.83% | 0.56% | 0.25% | 0.12% | 0.06% |

**Analysis of Extra Overheads of Prefilling.** The extra overhead of prefilling mainly comes from the prior estimation, which is measured by comparing the time (w/ and w/o RESA) of the last chunk and the entire prefilling. Given chunk size 1024 and the varying context length from 8k to 128k, the layer-wise latency (ms) of the last chunk and the entire prefilling of Llama-3.1-8B is shown in Table 3, with each test runs 100 times and averaged. We can see that the ratio of extra overhead almost gradually decreases and tends to stabilize with the expansion of context length. For the last chunk, the extra overhead is nearly stable at 5%. However, in practice we primarily concern the entire prefilling, whose extra overhead is negligible. This is because the longer the context, the more chunks need to be computed, while RESA only applies prior estimation to the last chunk.

**Analysis of Extra Overheads of Decoding.** The extra overhead of decoding mainly comes from the Online Aggregator, which merges the prior and real logits selected by SA. Given the same KV budget 500, we compare the layer-wise latency (ms) of each step w/ and

Table 4: The layer-wise decoding latency (ms) of RESA using Llama-3.1-8B-128k under varying context length.

| context length | 8k | 16k | 24k | 32k |
|---|---|---|---|---|
| w/o RESA | 1.92 | 2.81 | 3.68 | 4.28 |
| w/ RESA | 1.98 | 2.89 | 3.77 | 4.38 |
| extra overhead ratio | 3.13% | 2.85% | 2.45% | 2.34% |

w/o RESA. The results are shown in Table 4, with each test run 100 times and averaged. We can see that the extra overhead is less than 3.13% for each step. As the sequence increases, the additional overhead gradually decreases. This is because we keep the computational complexity of Online Aggregator the same as SA and all the extra computations of the Online Aggregator are lightweight, element-wise multiplication or addition, without any matrix multiplication.

**Analysis of Extra Overheads of Memory.** For a specific layer, the extra memory overhead mainly consists of four parts: 1. The prior logits with shape $(B, H, L, 1)$. 2. The output $O_{est}$ corresponds to the prior with shape $(B, H, 1, d)$. 3. The mean of queries $\mu_Q$ with shape $(B, H, 1, d)$. 4. The mean of keys $\mu_K$ with shape $(B, H, 1, d)$. $B, H, L, d$ denote batch size, number of head, sequence length and head dim. Considering that the original KV cache size is $(B, H, L, d)$, therefore, the ratio of extra memory overhead is: $r = \frac{1}{d} + \frac{3}{L}$. When $L$ is very large, $r \approx \frac{1}{d}$. Since a common configuration of $d$ is 128, 256 etc., the extra memory overhead is less than 1% and can be ignored.

## 6 CONCLUSION

In this paper, we propose RESA to enhance current sparse attention to improve model quality and efficiency. RESA fully explores the low-rank nature of attention logits matrix and leverages it to estimate a prior distribution for KV caches ignored by sparse attention. RESA then performs lightweight online aggregation to merge the prior and sparse attention's output. Experiments show that RESA improves model quality and efficiency compared to current sparse attention methods.

## 7 ACKNOWLEDGMENT

We thank the anonymous reviewers for their constructive suggestions. This research was partly supported by the Major Key Project of PCL under Grant PCL2024A05, the National Natural Science Foundation of China under Grants 62472127 and 62502119, and GuangDong Basic and Applied Basic Research Foundation under Grant 2023A1515110072.

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

# A APPENDIX

## A.1 MORE DETAILS OF RESA'S PRIOR LOSITS APPROXIMATION

We provide a clearer and more formal explanation of why our approximation is effective:

1. **The prior $\mu_Q K^T$ represents the mean of historical logits, which aligns closely with the principal SVD component.** For query $q_i$, given the true logit $z_i = q_i K^T$, our prior $\mu_Q K^T = (\frac{1}{L} \sum_{i=1}^{L} q_i) K^T = \frac{1}{L} \sum_{i=1}^{L} z_i$, which represents the mean value across the entire prefilling attention logits. Using this as the prior is because that we observed the historical logits of each token (i.e., a column of attention logits matrix) exhibit a concentrated structure: these logits mostly revolve around their averaged logit with relative small fluctuations (figures will be added in the final version). Thus, the corresponding elements in each row of the attention logits matrix are unlikely to change significantly (if there are significant changes, they will be captured by sparse attention). This leads to the rank-1 spike in attention logits. In such cases, when rows of attention logits matrix share strong similarities at most time, their mean vector is almost collinear with the principal singular vector. Thus, $\mu_Q K^T$ naturally aligns with the principal SVD component. However, note that the mean of $q_i$ here assumes that the attention logits matrix has no causal mask (i.e., a complete square matrix). If it has a causal mask (i.e., a lower triangular square matrix), then the more recent the token, the fewer the historical logits, which is more likely to cause fluctuations for prior estimation.

2. **The bias term $(q_i - \mu_Q) \cdot \mu_K^T$ enforces exact alignment of the true logit mean, which avoids the distortions when merging with sparse attention.** Given the true logits $q_i K^T$, the mean value of $q_i K^T$ over all keys is $q_i \mu_K^T$, where $\mu_K^T$ is the mean of all keys. However, considering our prior estimation $\mu_Q K^T$, its mean value is $\mu_Q \mu_K^T$, which is not aligning with the ground truth. Thus, our proposed bias term $(q_i - \mu_Q) \cdot \mu_K^T$ provides an exact correction such that the estimated logits and the true logits share the same mean. Aligning the mean of prior and ground truth eliminating global bias shifts, which is crucial to avoid systematic distortions of original attention. This is because softmax is very sensitive to relative, not absolute logits, so when aggregating prior and real logits selected by sparse attention, a large global bias will significantly change the distribution of attention scores. Thus, the bias term $(q_i - \mu_Q) \cdot \mu_K^T$ can significantly stabilize the approximation, and it is deterministic, training-free and can be adaptively adjusted through online query $q_i$.

3. **The remaining error is a second-order interaction of query/key deviations that is relative small.** For $q_i$, the error $\delta$ between true logits and our estimated logits is $\delta = q_i K^T - (\mu_Q K^T + (q_i - \mu_Q) \cdot \mu_K^T) = (q_i - \mu_Q) \cdot (K - \mu_K)^T$, which shows that $\delta$ is a bilinear interaction between query deviation and key deviation. Hence, it is second-order relative to the dominant mean-based structure. In Figure 6(b) of our paper, we visualized this error term and found some mean-reverting patterns. While there is no definitive hypothesis, we do believe there is a strong correlation between error and RoPE. Specifically, comparing attention logits with and without ROPE, we found that in some heads, RoPE does provide a near-periodic, relatively fixed bias, which we call the positional bias, and its trend closely resembles that of error. However, this is not universally applicable, and a quantitative relationship cannot be well established. We are now conducting further investigations to reveal the deeper relationship.

## A.2 MORE DETAILS OF SELECTING TYPICAL QUERY $\mu_Q$

In our paper, $\mu_Q$ is indeed defined as the average of all queries. First, we need to clarify that such a determination actually utilizes the mean of historical logits as the prior (historical logits often exhibit mean regression characteristics). Second, for other methods of choosing $\mu_Q$, such as directly selecting the last query, we experimentally demonstrate that their performance is inferior to the average of the queries. Some results of Llama-3.2-3B is shown in Table A.2.

Table 5: The performance of Llama3.2-3B with different typical query $\mu_Q$.

| Llama3.2-3B-8K | Niah3 | MK1 | MK2 | MK3 | MV | MQ |
|---|---|---|---|---|---|---|
| ArkVale-mean-q | 97 | 98 | 79 | 6 | 88 | 94 |
| ArkVale-last-q | 89 | 97 | 79 | 2 | 82 | 92.75 |
| Quest-mean-q | 84 | 98 | 82 | 5 | 84.25 | 93 |
| Quest-last-q | 74 | 94 | 78 | 3 | 74.5 | 89.5 |

### A.3 More Details of Motivations

First of all, for Llama-3.1-8B, we provide the accumulate ratio of singular values for more layers and heads. Then we provide the ratio of the principal singular value with different context length. Finally, we show the linear scaling property of principal singular values for different layers and heads. It is easy to observe that the logits matrix has a significant low rank property, and the main singular value is very important. This also fully reveals there are substantial redundancy in the logits.

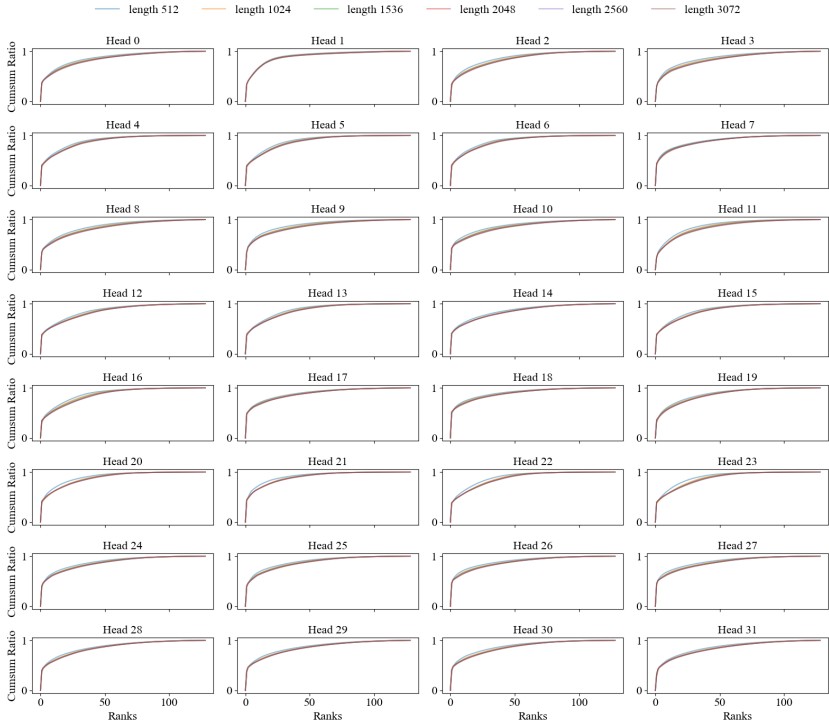

Figure 8: **The accumulate ratio of singular values for layer 8.**

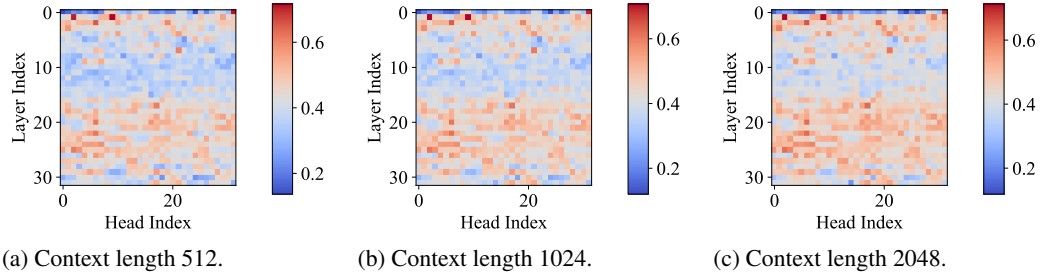

(a) Context length 512.   (b) Context length 1024.   (c) Context length 2048.

Figure 9: **The ratio of principal singular value under different context length.**

Table 6: Llama3.1-8b quality verification of RULER with Quest under different budget.

| llama3.1-8b | Niah1 | Niah2 | Niah3 | MK1 | MK2 | MK3 | MV | MQ | VT | CWE | FWE | QA1 | QA2 |
|---|---|---|---|---|---|---|---|---|---|---|---|---|---|
| quest-5% | 100 | 100 | 100 | 100 | 97 | 50 | 97.75 | 99 | 94.2 | 49.9 | 78.67 | 88 | 57 |
| quest-5%+RESA | 100 | 100 | 100 | 100 | 96 | 69 | 99.5 | 99.5 | 96.2 | 49.7 | 79.67 | 88 | 57 |
| quest-7.5% | 100 | 100 | 100 | 100 | 100 | 74 | 98 | 99 | 95.2 | 65.5 | 78 | 88 | 55 |
| quest-7.5%+RESA | 100 | 100 | 100 | 100 | 100 | 82 | 99.75 | 99.5 | 95.2 | 65.5 | 78 | 88 | 57 |
| quest-10% | 100 | 100 | 100 | 100 | 99 | 86 | 99 | 99.25 | 95.6 | 73.8 | 76.67 | 88 | 56 |
| quest-10%+RESA | 100 | 100 | 100 | 100 | 99 | 91 | 99.75 | 99.5 | 95.6 | 74.2 | 76.67 | 88 | 57 |

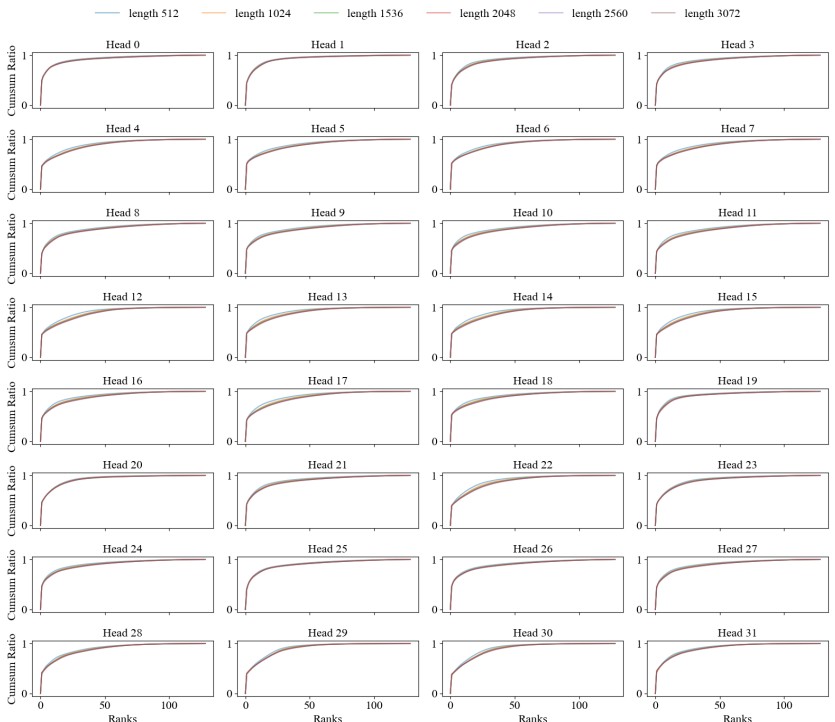

Figure 10: **The accumulate ratio of singular values for layer 16.**

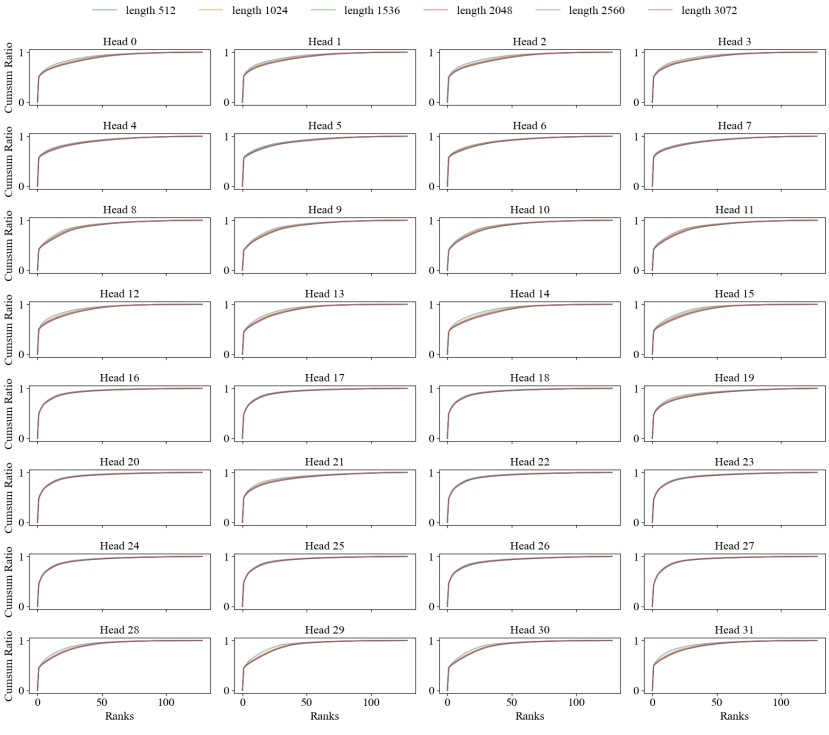

Figure 11: **The accumulate ratio of singular values for layer 24.**

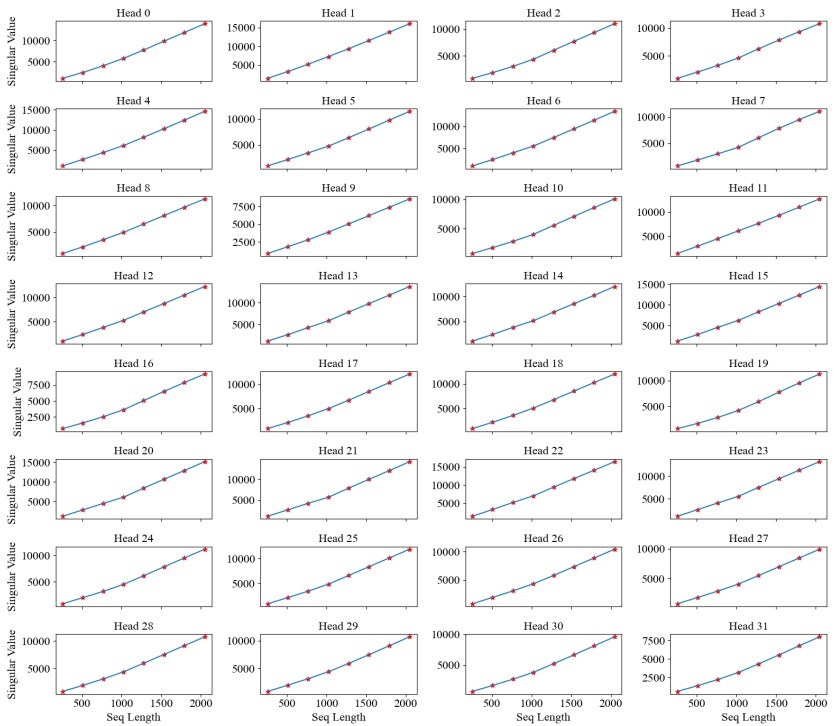

Figure 12: **The ratio of principal singular values for layer 8.**

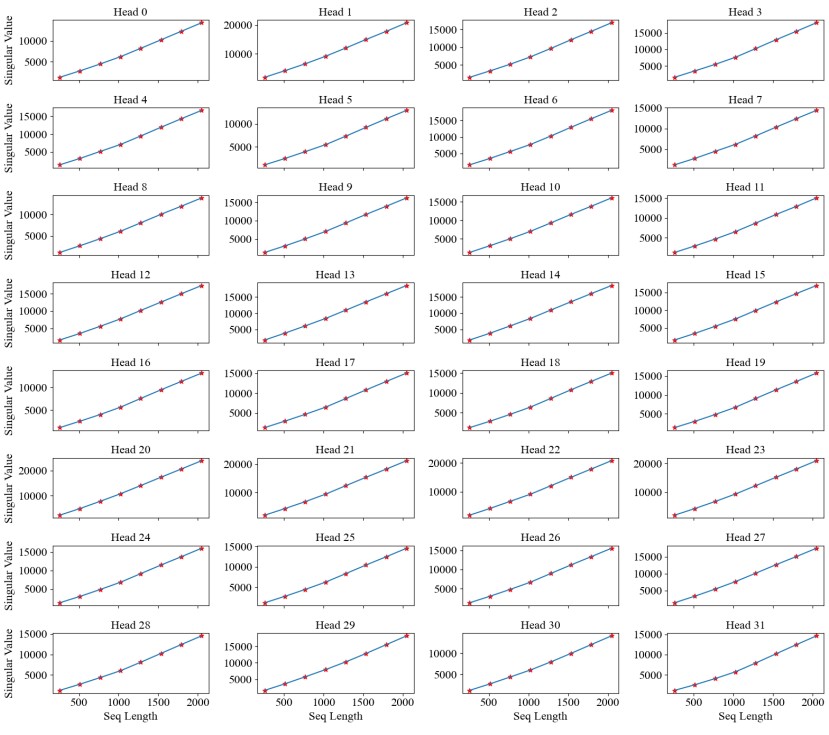

Figure 13: **The ratio of principal singular values for layer 16.**

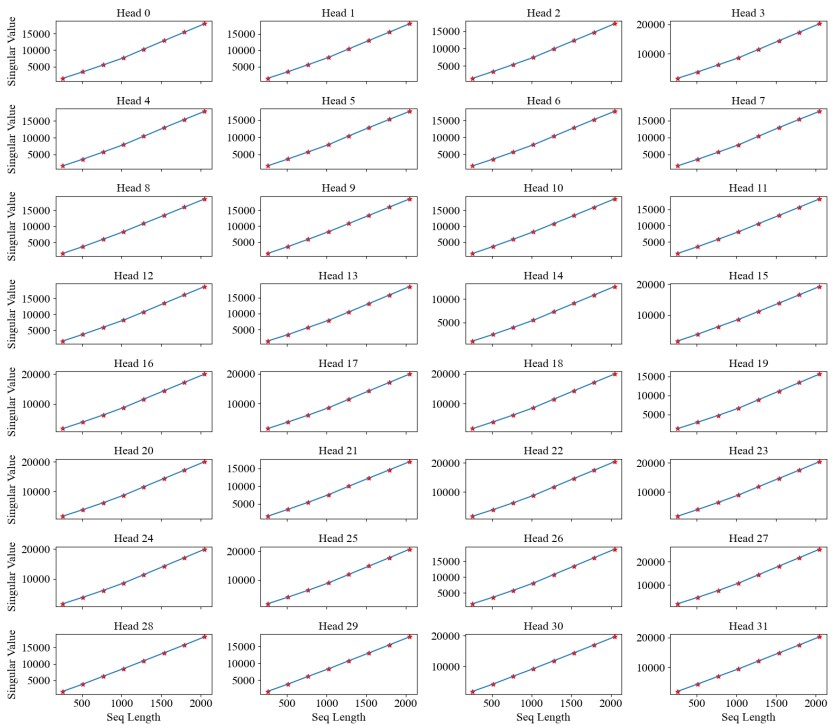

Figure 14: **The ratio of principal singular values for layer 24.**

A.4    MORE DETAILS OF ACCURACY EVALUATION

We show more results of RULER and LoongBench with different configurations.

Table 7: Llama3.1-8b quality verification of RULER with Arkvale under different budget.

| llama3.1-8b | Niah1 | Niah2 | Niah3 | MK1 | MK2 | MK3 | MV | MQ | VT | CWE | FWE | QA1 | QA2 |
|---|---|---|---|---|---|---|---|---|---|---|---|---|---|
| arkvale-5% | 100 | 100 | 100 | 100 | 97 | 60 | 97.5 | 99.5 | 97.8 | 56.1 | 79 | 89 | 54 |
| arkvale-5%+RESA | 100 | 100 | 100 | 100 | 97 | 66 | 98.5 | 99.5 | 98.2 | 56.2 | 80 | 89 | 54 |
| arkvale-7.5% | 100 | 100 | 100 | 100 | 98 | 78 | 99.5 | 99.25 | 97.2 | 67.1 | 78.67 | 89 | 55 |
| arkvale-7.5%+RESA | 100 | 100 | 100 | 100 | 98 | 84 | 99.75 | 99.5 | 98.8 | 71.4 | 80.33 | 89 | 57 |
| arkvale-10% | 100 | 100 | 100 | 100 | 99 | 88 | 99.25 | 99.5 | 98.6 | 73.6 | 79.33 | 89 | 55 |
| arkvale-10%+RESA | 100 | 100 | 100 | 100 | 99 | 95 | 99.5 | 99.5 | 98.6 | 73.6 | 79.33 | 89 | 57 |

Table 8: Llama3.2-3b quality verification of RULER with Quest under different budget.

| llama3.2-3b | Niah1 | Niah2 | Niah3 | MK1 | MK2 | MK3 | MV | MQ | VT | CWE | FWE | QA1 | QA2 |
|---|---|---|---|---|---|---|---|---|---|---|---|---|---|
| quest-5% | 100 | 93 | 94 | 98 | 92 | 11 | 87 | 96.25 | 68.8 | 25.4 | 71.67 | 64 | 45 |
| quest-5%+RESA | 100 | 93 | 94 | 99 | 94 | 12 | 92.75 | 97 | 68.8 | 25.4 | 71.67 | 64 | 49 |
| quest-7.5% | 100 | 95 | 97 | 100 | 94 | 20 | 92.75 | 95.5 | 68.6 | 30.9 | 74 | 65 | 48 |
| quest-7.5%+RESA | 100 | 95 | 97 | 100 | 95 | 28 | 94.5 | 97.25 | 68.6 | 30.9 | 74 | 65 | 48 |
| quest-10% | 100 | 95 | 99 | 100 | 96 | 34 | 92 | 97.75 | 69.6 | 32 | 76 | 63 | 48 |
| quest-10%+RESA | 100 | 95 | 99 | 100 | 97 | 38 | 94.75 | 97.75 | 69.6 | 32 | 76 | 64 | 49 |

Table 9: Llama3.2-3b quality verification of RULER with Arkvale under different budget.

| llama3.2-3b | Niah1 | Niah2 | Niah3 | MK1 | MK2 | MK3 | MV | MQ | VT | CWE | FWE | QA1 | QA2 |
|---|---|---|---|---|---|---|---|---|---|---|---|---|---|
| arkvale-5% | 100 | 94 | 99 | 98 | 90 | 14 | 91 | 95.5 | 70.4 | 26.4 | 69.67 | 62 | 49 |
| arkvale-5%+RESA | 100 | 94 | 100 | 98 | 94 | 16 | 94.5 | 96.25 | 70.4 | 26.4 | 69.67 | 63 | 49 |
| arkvale-7.5% | 100 | 95 | 100 | 99 | 95 | 26 | 93 | 97 | 71.6 | 31.8 | 72.67 | 61 | 49 |
| arkvale-7.5%+RESA | 100 | 95 | 100 | 99 | 95 | 36 | 94.25 | 97 | 71.6 | 31.8 | 72.67 | 62 | 49 |
| arkvale-10% | 100 | 95 | 100 | 100 | 97 | 44 | 93.25 | 97.75 | 70.2 | 31.4 | 75 | 63 | 48 |
| arkvale-10%+RESA | 100 | 95 | 100 | 100 | 97 | 48 | 94.5 | 97.75 | 70.2 | 31.4 | 75 | 63 | 49 |

Table 10: Llama3.1-8b quality verification of Longbench with 0-4k context length.

| llama3.1-8b | HQA | 2WMQA | QAs | MFQA | GR | MN | Tre | TQA | PC | PR |
|---|---|---|---|---|---|---|---|---|---|---|
| quest-5% | 4.6 | 6.89 | 11.15 | 17.69 | 26.26 | 25.37 | 50 | 91.82 | 20.47 | 93.47 |
| quest+RESA-5% | 4.7 | 6.91 | 11.49 | 17.92 | 26.31 | 25.73 | 50 | 91.89 | 20.59 | 97.28 |
| arkvale-5% | 4.14 | 7.06 | 11.18 | 16.21 | 26.86 | 25.42 | 50 | 90.1 | 21.27 | 90.22 |
| arkvale+RESA-5% | 4.59 | 7.69 | 11.28 | 16.48 | 26.99 | 26.24 | 50 | 90.1 | 26.95 | 92.79 |
| topk-5% | 6.71 | 9.35 | 12.1 | 20.96 | 26.85 | 25.19 | 50 | 95.24 | 21.61 | 94.14 |
| topk+RESA-5% | 6.79 | 9.65 | 12.1 | 20.98 | 26.85 | 25.23 | 50 | 95.52 | 23.89 | 98.89 |

Table 11: Llama3.2-3b quality verification of Longbench with 0-4k context length.

| llama3.2-3b | HQA | 2WMQA | QAs | MFQA | GR | MN | Tre | TQA | PC | PR |
|---|---|---|---|---|---|---|---|---|---|---|
| quest-5% | 35.18 | 23.74 | 45.39 | 47.89 | 24.97 | 24.92 | 45.83 | 81.9 | 18.75 | 88.33 |
| quest+RESA-5% | 35.67 | 23.91 | 45.39 | 48.15 | 25.77 | 25.36 | 45.83 | 81.94 | 18.75 | 90.52 |
| arkvale-5% | 35.47 | 25.75 | 41.51 | 45.41 | 26.45 | 41.67 | 82.02 | 18.75 | 89.69 | |
| arkvale+RESA-5% | 35.87 | 29.13 | 44.09 | 47.73 | 26.46 | 25.51 | 41.67 | 82.26 | 18.75 | 89.91 |
| topk-5% | 38.16 | 32.6 | 45.51 | 47.72 | 27.73 | 26.64 | 45.83 | 84.02 | 18.75 | 96.37 |
| topk+RESA-5% | 43.7 | 32.81 | 45.51 | 48.77 | 27.73 | 26.64 | 45.83 | 84.02 | 18.75 | 97.37 |

Table 12: Mistral-7B quality verification of Longbench with 0-4k context length.

| Mistral-7B | HQA | 2WMQA | QAs | MFQA | GR | MN | Tre | TQA | PC | PR |
|---|---|---|---|---|---|---|---|---|---|---|
| quest-5% | 29.81 | 36.38 | 9.8 | 44.68 | 23.66 | 23.46 | 37.5 | 95.71 | 0 | 100 |
| quest+RESA-5% | 38.54 | 36.52 | 10.95 | 46.61 | 23.66 | 24.2 | 37.5 | 95.71 | 0 | 100 |
| arkvale-5% | 32.41 | 32.31 | 12.19 | 43.04 | 22.94 | 22.92 | 41.67 | 94.76 | 0 | 100 |
| arkvale+RESA-5% | 40.33 | 32.83 | 12.9 | 47.07 | 23.28 | 25.07 | 41.67 | 94.76 | 0 | 100 |
| topk-5% | 26.72 | 31.79 | 13.26 | 39.84 | 23.5 | 24.04 | 41.67 | 91.43 | 0 | 100 |
| topk+RESA-5% | 33.94 | 34.23 | 13.64 | 44.88 | 23.5 | 24.23 | 41.67 | 92.86 | 0 | 100 |

Table 13: LWM-Text-7B quality verification of Longbench with 0-4k context length.

| LWM-Text-7B | HQA | 2WMQA | QAs | MFQA | GR | MN | Tre | TQA | PC | PR |
|---|---|---|---|---|---|---|---|---|---|---|
| quest-5% | 3.16 | 3.55 | 6.92 | 16.07 | 22.4 | 16.58 | 37.5 | 94.15 | 0.74 | 4.02 |
| quest+RESA-5% | 3.51 | 4.16 | 7.14 | 18.87 | 22.57 | 17.33 | 37.5 | 96.19 | 3.44 | 5.19 |
| arkvale-5% | 2.98 | 3.74 | 7.5 | 13.19 | 21.71 | 13.53 | 33.33 | 94.15 | 0.6 | 6.01 |
| arkvale+RESA-5% | 3.52 | 4.17 | 7.64 | 17.54 | 21.79 | 17.59 | 37.5 | 96.19 | 3.47 | 9.23 |
| topk-5% | 3.49 | 4.19 | 7.58 | 15.21 | 22.34 | 19.98 | 41.67 | 95.24 | 3.12 | 7.3 |
| topk+RESA-5% | 3.52 | 4.37 | 8.78 | 15.67 | 22.39 | 19.98 | 41.67 | 97.62 | 3.12 | 7.3 |

Table 14: Llama3.1-8b quality verification of Longbench with 8k+ context length.

| llama3.1-8b | HQA | 2WMQA | QAs | MFQA | GR | MN | Tre | TQA | PC | PR |
|---|---|---|---|---|---|---|---|---|---|---|
| quest-5% | 5.6 | 4.12 | 8.98 | 22.96 | 23.41 | 20.41 | 71.05 | 94.38 | 10.24 | 91.81 |
| quert+RESA-5% | 6.2 | 4.23 | 9.75 | 23.05 | 23.74 | 20.81 | 71.05 | 94.38 | 12.86 | 93.46 |
| arkvale-5% | 5.35 | 4.15 | 10.52 | 21.88 | 22.89 | 19.64 | 71.05 | 94.38 | 12.44 | 92.65 |
| arkvale+RESA-5% | 10.03 | 4.56 | 13.12 | 23.75 | 23.6 | 21.22 | 71.05 | 94.38 | 13.14 | 92.66 |
| topk-5% | 5.02 | 4.14 | 11.69 | 21.88 | 23.48 | 20.32 | 71.05 | 94.38 | 9.8 | 93.42 |
| topk+RESA-5% | 5.37 | 4.95 | 11.78 | 22.21 | 23.61 | 20.26 | 71.05 | 94.38 | 13.79 | 93.89 |

Table 15: Llama3.2-3b quality verification of Longbench with 8k+ context length.

| llama3.2-3b | HQA | 2WMQA | QAs | MFQA | GR | MN | Tre | TQA | PC | PR |
|---|---|---|---|---|---|---|---|---|---|---|
| quest-5% | 26.24 | 13.97 | 29.17 | 56.21 | 23.11 | 21.18 | 71.05 | 90.67 | 3.03 | 85.08 |
| quert+RESA-5% | 28.09 | 14.55 | 36.52 | 56.88 | 23.33 | 21.18 | 71.05 | 92.67 | 3.03 | 88.71 |
| arkvale-5% | 25.68 | 14.18 | 31.06 | 55.84 | 23.69 | 19.92 | 71.05 | 90.67 | 3.03 | 85.23 |
| arkvale+RESA-5% | 27.17 | 14.89 | 35.72 | 57.88 | 23.71 | 21.74 | 71.05 | 90.67 | 3.03 | 87.05 |
| topk-5% | 31.35 | 17.33 | 29.17 | 57.78 | 23.13 | 19.35 | 71.05 | 90.67 | 3.03 | 88.48 |
| topk+RESA-5% | 33.26 | 18.25 | 31.51 | 57.78 | 23.83 | 21.37 | 71.05 | 92.67 | 3.03 | 88.48 |

Table 16: Mistral-7B quality verification of Longbench with 8k+ context length.

| Mistral-7B | HQA | 2WMQA | QAs | MFQA | GR | MN | Tre | TQA | PC | PR |
|---|---|---|---|---|---|---|---|---|---|---|
| quest-5% | 29.81 | 7.31 | 10.01 | 65.84 | 19.88 | 20.26 | 78.95 | 89.3 | 0 | 96.77 |
| quert+RESA-5% | 30.41 | 9.75 | 10.01 | 65.84 | 19.92 | 20.26 | 78.95 | 89.3 | 0.11 | 96.77 |
| arkvale-5% | 30.19 | 7.36 | 10.41 | 66.31 | 19.66 | 20.8 | 78.95 | 92.63 | 0 | 96.77 |
| arkvale+RESA-5% | 30.19 | 12.31 | 10.58 | 67.02 | 19.66 | 20.8 | 78.95 | 92.63 | 0 | 96.77 |
| topk-5% | 29.77 | 12.09 | 9.93 | 69.78 | 19.07 | 20.02 | 78.95 | 90.63 | 0 | 96.77 |
| topk+RESA-5% | 35.86 | 16.03 | 10.08 | 69.92 | 19.26 | 20.29 | 78.95 | 90.63 | 0 | 96.77 |

Table 17: LWM-Text-7B quality verification of Longbench with 8k+ context length.

| LWM-Text-7B | HQA | 2WMQA | QAs | MFQA | GR | MN | Tre | TQA | PC | PR |
|---|---|---|---|---|---|---|---|---|---|---|
| quest-5% | 0.93 | 0.07 | 0.38 | 8.35 | 3.77 | 4.14 | 13.16 | 8 | 0 | 1.01 |
| quert+RESA-5% | 0.97 | 0.24 | 0.38 | 8.55 | 3.77 | 4.17 | 13.16 | 12 | 0 | 1.35 |
| arkvale-5% | 1.54 | 0.37 | 0.38 | 4.9 | 5.27 | 13.16 | 12 | 0 | 1.81 |
| arkvale+RESA-5% | 1.54 | 0.42 | 0.38 | 10.12 | 4.95 | 5.79 | 13.16 | 12 | 0 | 2.54 |
| topk-5% | 1.04 | 0.97 | 0.39 | 10.95 | 4.83 | 5.47 | 15.79 | 12 | 0 | 1.53 |
| topk+RESA-5% | 1.53 | 0.97 | 0.68 | 10.95 | 6.27 | 6.16 | 15.97 | 12 | 0 | 2.42 |

## A.5 MORE DETAILS OF ERROR ANALYSIS

We show some randomly selected tokens and their error between real logits and the estimated logits given by RESA.

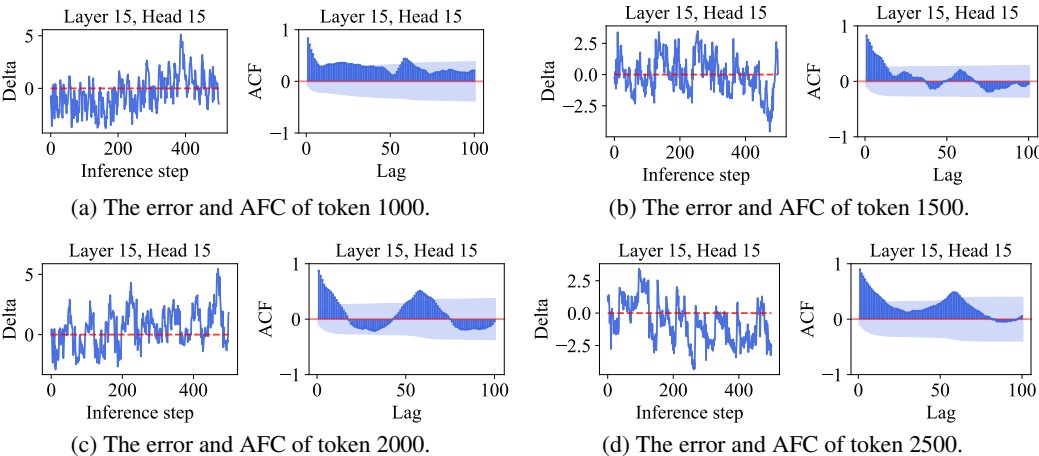

(a) The error and AFC of token 1000.

(b) The error and AFC of token 1500.

(c) The error and AFC of token 2000.

(d) The error and AFC of token 2500.

Figure 15: **The error between real logits and estimated logits (randomly selected tokens.)**

