# OpenReview forum: "RESA: Bringing Back What Sparse Attention Ignores with Residual Estimation"
_ICLR.cc/2026/Conference — ICLR 2026 Poster_

### Official Review · Reviewer_5DL8 · 2025-10-24

**Soundness:** 3
**Presentation:** 3
**Contribution:** 3
**Rating:** 6
**Confidence:** 4

**Summary:**

This paper proposes RESA (Residual Estimation for Sparse Attention), a novel, training-free framework to improve the accuracy of sparse attention (SA) during LLM inference. The authors argue that standard SA methods, which select a small subset of KVs and ignore the rest, suffer from accuracy degradation by effectively zeroing out the contribution of unselected tokens.

RESA's core idea is to *estimate* the contribution of these ignored KVs rather than discarding them. This is motivated by the empirical observation that the attention logit matrix ($QK^T$) is inherently low-rank and is dominated by its principal (rank-1) singular value, which accounts for a significant portion (40-50%) of the total energy.

Instead of performing a costly SVD, RESA proposes a lightweight approximation for this rank-1 component. This approximation is used by a "Prior Estimator" to compute a global "prior output" ($O_{Est}$) once at the end of the prefill stage. Then, at each decoding step, an "Online Aggregator" merges this pre-computed $O_{Est}$ with the output of a standard SA method (e.g., Quest, ArkVale). The paper presents a key algorithmic contribution in the form of a "delta merging" formula (Eq 5), which allows this aggregation to be performed in $O(|\mathcal{I}|)$ time—the same asymptotic complexity as the original SA method—by cleverly rescaling and combining the pre-computed and online components.

Experiments across multiple models (Llama 3.1, Mistral) and benchmarks (RULER, LongBench) demonstrate that RESA consistently improves the accuracy of existing SA methods, or alternatively, allows for a significant reduction in the KV budget (up to 33.2%) while maintaining the same accuracy.

**Strengths:**

1.  **Novel and Valuable Perspective:** The core idea of "residual estimation" is a significant conceptual strength. It reframes the problem of information loss in SA and provides a new, orthogonal axis for improvement.
2.  **Strong Empirical Motivation:** The SVD analysis in Section 3.2 (Figure 2) is clear, convincing, and provides a strong, data-driven justification for why a low-rank (specifically, rank-1) approximation is a sensible approach.
3.  **Algorithmically Efficient Design:** The "delta merging" technique in Section 4.3 is a key technical contribution. Deriving a formula (Eq 5) that correctly merges the pre-computed prior with the online sparse logits, all while maintaining the $O(|\mathcal{I}|)$ asymptotic complexity of vanilla SA, is non-trivial and essential for the method's practicality.
4.  **Training-Free and General:** The fact that RESA requires no re-training and can be applied as a wrapper around existing SA methods (Quest, ArkVale, etc.) makes it highly general and easy to adopt.
5.  **Consistent Positive Results:** The experiments compellingly show that RESA provides a clear benefit, either by improving accuracy at a fixed budget (Tables 1, 2) or by enabling significant budget reduction (up to 33.2%) for a fixed accuracy target (Figure 7b).
Markdow

**Weaknesses:**

1.  **Missing Derivation for the Core Approximation (Eq 2):** This is the paper's most significant weakness. The "Prior Estimator" is built entirely on the approximation $q_{i}\cdot K^{T}\approx\mu_{Q}\cdot K^{T}+(q_{i}-\mu_{Q})\cdot\mu_{K}^{T}$. The paper provides no derivation for this formula or theoretical justification for why it should approximate the rank-1 SVD component $\sigma_{1}u_{1,i}v_{1}^{T}$. It is simply presented, and its validity is only asserted empirically (Figure 4). This leaves a critical gap in the paper's technical story.
2.  **Speculative RoPE Analysis:** The connection made in Section 5.2 between the $\mu_Q$/$\mu_K$ terms and the "low-frequency semantic carriers" of RoPE (Figure 5b) is interesting but highly speculative. It reads as a post-hoc justification rather than a core part of the method's design, and it's not strictly necessary to validate the method's performance.
3.  **Ambiguity in "Overhead":** The paper claims "without extra overheads" and "maintains the same complexity $O(|\mathcal{I}|)$ as SA." While asymptotically true for the *decoding step*, this glosses over:
    * (a) The $O(L \cdot d)$ computation and $O(L)$ softmax for $O_{Est}$ at the end of prefill (though this one-time cost is likely amortized and acceptable).
    * (b) The *constant factor* overhead of the delta-merge (Eq 5) at *every* decoding step. The paper shows throughput gains *after* reducing the budget (Fig 7b), but it never shows a direct, per-token latency comparison of SA vs. SA+RESA at the *same* budget.

**Questions:**

1.  Could the authors please provide a derivation or at least a more formal justification for Equation 2? How is the expression $\mu_{Q}\cdot K^{T}+(q_{i}-\mu_{Q})\cdot\mu_{K}^{T}$ formally linked to the principal SVD component $\sigma_{1}u_{1,i}v_{1}^{T}$?
2.  To clarify the true overhead, what is the measured, per-token wall-clock latency (at the *same* KV budget, e.g., 5%) for a baseline like Quest versus Quest+RESA? This would quantify the constant-factor overhead of the Online Aggregator.
3.  What is the additional *memory* overhead of RESA? The $O_{Est}$ vector (size $d$) and the prior scores $S$ (size $L$) must be stored for every head and every layer, correct? How does this extra storage compare to the size of the KV cache itself?

---

> ### Author Response · Authors · 2025-11-22
> **Response to Reviewer 5DL8**
>
> We thank the reviewer for raising concerns, we will answer questions unrelated to efficiency below.
> Questions about efficiency can be seen in the **Response to All Reviewers**.
>
> **Q1:**
> Could the authors please provide a derivation or at least a more formal justification for Equation 2?
> How is the expression $\mu_QK^T + (q_i-\mu_Q)\cdot \mu_K^T$ formally linked to the principal SVD component $\sigma_1u_{1,i}v_1^T$?
>
> **A1:**
> We provide a clearer and more formal explanation of why our approximation is effective from three aspects:
>
> *1. The prior $\mathbf{\mu_Q K^T}$ represents the mean of historical logits, which aligns closely with the principal SVD component.*
> For query $q_i$, given the true logit $z_i=q_iK^T$,
> our prior $\mu_QK^T=(\frac{1}{L}\sum_{i=1}^Lq_i)K^T=\frac{1}{L}\sum_{i=1}^Lz_i$, which represents *the mean value across the entire prefilling attention logits*.
> Using this as the prior is because that we observed the historical logits of each token (i.e., a column of attention logits matrix) exhibit a concentrated structure: *these logits mostly revolve around their averaged logit with relative small fluctuations (figures will be added in the final version).*
> Thus, the corresponding elements in each row of the attention logits matrix are unlikely to change significantly (if there are significant changes, they will be captured by sparse attention).
> This leads to the *rank-1 spike* in attention logits.
> In such cases, when rows of attention logits matrix share strong similarities at most time, their mean vector is almost collinear with the principal singular vector.
> Thus, $\mu_QK^T$ naturally aligns with the principal SVD component.
> However, note that the mean of $q_i$ here assumes that the attention logits matrix has no causal mask (i.e., a complete square matrix).
> If it has a causal mask (i.e., a lower triangular square matrix), then the more recent the token, the fewer the historical logits, which is more likely to cause fluctuations for prior estimation.
>
> *2. The bias term $(q_i-\mu_Q)\cdot \mu_K^T$ enforces exact alignment of the true logit mean, which avoids the distortions when merging with sparse attention.*
> Given the true logits $q_iK^T$, the mean value of $q_iK^T$ over all keys is $q_i\mu_K^T$, where $\mu_K^T$ is the mean of all keys.
> However, considering our prior estimation $\mu_QK^T$, its mean value is $\mu_Q\mu_K^T$, which is not aligning with the ground truth.
> Thus, our proposed bias term $(q_i-\mu_Q)\cdot \mu_K^T$ provides *an exact correction such that the estimated logits and the true logits share the same mean.*
> *Aligning the mean of prior and ground truth eliminating global bias shifts, which is crucial to avoid systematic distortions of original attention.*
> This is because softmax is very sensitive to relative, not absolute logits, so when aggregating prior and real logits selected by sparse attention, a large global bias will significantly change the distribution of attention scores.
> Thus, the bias term $(q_i-\mu_Q)\cdot \mu_K^T$ can significantly stabilize the approximation, and it is deterministic, training-free and can be adaptively adjusted through online query $q_i$.
>
> *3. The remaining error is a second-order interaction of query/key deviations that is relative small.*
> For $q_i$, the error $\delta$ between true logits and our estimated logits is $\delta=q_iK^T-(\mu_QK^T + (q_i-\mu_Q)\cdot \mu_K^T)=(q_i-\mu_Q)\cdot(K-\mu_K)^T$, which shows that $\delta$ is a bilinear interaction between query deviation and key deviation.
> Hence, it is second-order relative to the dominant mean-based structure.
> In Figure 6(b) of our paper, we visualized this error term and find some mean-reverting patterns.
> Furthermore, in our response to **Reviewer 2Wxj** (A4), we also mentioned a hypothesis that might be related to the fixed bias introduced by RoPE.
> This is an interesting area we are currently exploring.
> (This answer is similar to the response to the **Reviewer WqTC** A1.)
>
>
> **Q2:**
> To clarify the true overhead, what is the measured, per-token wall-clock latency (at the same KV budget, e.g., 5\%) for a baseline like Quest versus Quest+RESA?
> This would quantify the constant-factor overhead of the Online Aggregator.
>
> **A2:**
> For more detailed evaluation about efficiency, please refer to the **Response to All Reviewers** (the part of decoding overhead).
>
>
> **Q3:**
> What is the additional memory overhead of RESA?
> The $O_{Est}$ vector (size $d$) and the prior scores $S$ (size $L$) must be stored for every head and every layer, correct?
> How does this extra storage compare to the size of the KV cache itself?
>
> **A3:**
> For more detailed evaluation about efficiency, please refer to the **Response to All Reviewers** (the part of memory overhead).

---

### Official Review · Reviewer_WmEf · 2025-10-30

**Soundness:** 3
**Presentation:** 3
**Contribution:** 2
**Rating:** 6
**Confidence:** 4

**Summary:**

This paper presents a novel and well-motivated approach to improving sparse attention (SA) mechanisms by leveraging the low-rank nature of attention logits. The proposed method, RESA, is training-free and introduces a lightweight residual estimation framework that compensates for the contributions of unselected key-value pairs. The idea is innovative and supported by solid theoretical and empirical analysis. However, the evaluation could be strengthened by more comprehensive comparisons and deeper ablations to better understand the method's generality and limitations.

**Strengths:**

- The paper provides a fresh perspective on the redundancy in attention logits due to their inherent low-rank structure, supported by singular value decomposition (SVD) analysis and visualizations. This offers a principled foundation for residual estimation in sparse attention.
- RESA is a training-free framework that effectively captures low-rank priors during prefilling and integrates them during decoding via a lightweight online aggregation mechanism. This makes it practical and easy to deploy.
- The method demonstrates consistent improvements across multiple models and benchmarks (RULER and LongBench), with significant gains (up to 26% on certain tasks) under the same KV budget.
- The authors further interpret RESA’s mechanism as a form of attention bias for knowledge injection, opening up interesting directions for training-time optimizations.

**Weaknesses:**

1. **Comparison with Recent SA Methods**:
   The paper primarily compares against Quest, ArkVale, and an Ideal Top-K baseline. However, several recent and relevant SA methods such as SnapKV, HeadKV, PSA, and Ada-KV are not included in the main evaluation. It would be beneficial to see how RESA performs when integrated with these more recent approaches, especially on the same benchmarks.

2. **Efficiency Evaluation**:
   While the method claims to be lightweight, there is no concrete reporting of prefilling time, inference latency, or memory footprint. Including such metrics would help readers better assess the practical overhead of RESA, especially in resource-constrained settings.

3. **Evaluation on Complex Tasks**:
   The current experiments focus heavily on retrieval and simple QA tasks. To better demonstrate the robustness of RESA, we suggest evaluating on more challenging benchmarks such as InfiniteBench or complex reasoning tasks from lm-eval, which involve few-shot learning, long-context reasoning, or multi-step inference.

4. **Prefilling Overhead**:
   The authors mention that the prior estimation is computed once at the end of prefilling, but the computational and memory cost of this step is not quantified. A brief analysis of the prefilling stage overhead would help clarify the trade-offs.

5. **Ablation Studies**:
   Although a hyperparameter \( \lambda \) is introduced to control the prior's influence, more thorough ablations are needed. For instance:
   - How does the choice of rank-1 approximation affect performance? Would a higher-rank prior help?
   - Are there alternative weighting schemes (e.g., recency-based) that could improve the prior?
   - How sensitive are the scaling factors \( \alpha \) and \( \beta \) in Eq. 5?
   These experiments would provide deeper insights into the design choices and their impact.

**Questions:**

See the weaknesses section for details

---

> ### Author Response · Authors · 2025-11-22
> **Response to Reviewer WmEf**
>
> We thank the reviewer for raising concerns, we will answer questions unrelated to efficiency below.
> Questions about efficiency can be seen in the **Response to All Reviewers**.
>
> **Q1:**
> The paper primarily compares against Quest, ArkVale, and an Ideal Top-K baseline.
> However, several recent and relevant SA methods such as HeadKV, PSA, and Ada-KV are not included in the main evaluation.
> It would be beneficial to see how RESA performs when integrated with these more recent approaches, especially on the same benchmarks.
>
> **A1:**
> The efficiency experiments include a comparison between PSA and Ada-KV.
> The comparison primarily focuses on how RESA accelerates by reducing I/O and computational overhead by decreasing the KV budget while maintaining accuracy.
> We didn't compare them for accuracy because *PSA and Ada-KV focus on how much budget to allocate, which is orthogonal to RESA's perspective.*
> For the accuracy experiments, RESA is mainly compared with various sparse attention algorithms, because they focus on *how to accurately select the most relevant KVs, while RESA estimates how to utilize unselected KVs*; their perspectives are complementary.
>
>
> **Q2:**
> While the method claims to be lightweight, there is no concrete reporting of prefilling time, inference latency, or memory footprint.
> Including such metrics would help readers better assess the practical overhead of RESA, especially in resource-constrained settings.
>
> **A2:**
> For more detailed evaluation about efficiency, please refer to the **Response to All Reviewers**.
>
>
> **Q3:**
> The current experiments focus heavily on retrieval and simple QA tasks.
> To better demonstrate the robustness of RESA, we suggest evaluating on more challenging benchmarks such as InfiniteBench or complex reasoning tasks from lm-eval, which involve few-shot learning, long-context reasoning, or multi-step inference.
>
> **A3:**
> We are trying more experiments and will add the necessary experiments in the final version.
>
>
> **Q4:**
> The authors mention that the prior estimation is computed once at the end of prefilling, but the computational and memory cost of this step is not quantified. A brief analysis of the prefilling stage overhead would help clarify the trade-offs.
>
> **A4:**
> For more detailed evaluation about efficiency, please refer to the **Response to All Reviewers** (the part of prefilling and memory overhead).
>
>
> **Q5:**
> Although a hyperparameter ($\lambda$) is introduced to control the prior's influence, more thorough ablations are needed.
> For instance:
> (i) How does the choice of rank-1 approximation affect performance? Would a higher-rank prior help?
> (ii) Are there alternative weighting schemes (e.g., recency-based) that could improve the prior?
> (iii)How sensitive are the scaling factors ($\alpha$) and ($\beta$) in Eq. 5?
>
> **A5:**
> (i) Theoretically, more ranks are always better, but since we don't want to perform SVD online due to its huge overhead, we can only do it in a training-free way.
> Currently, the only training-free method we've found is an approximation of rank-1.
>
> (ii) Our understanding of the weighting in the problem refers to weighting all queries to obtain $\mu_Q$.
> First, we need to clarify that using the mean of all queries as $\mu_Q$ actually utilizes the mean of historical logits as the prior (historical logits often exhibit mean regression characteristics).
> We provided more details and support in our response to **Reviewer WqTC** (A1) or **Reviewer 5DL8** (A1).
> Second, we have tried some other weighting methods, such as one-hot weighting, which selects only one query as the value of $\mu_Q$ to calculate the prior; however, this introduces a large selection space (e.g., if the context length is $L$, then there are $L$ selections), making it very inconvenient.
> Other methods like exponential smoothing weighting (i.e., $\mu_Q = q_L + w*q_{L-1} + ... + w^{L-1}*q_1$) introduces new hyperparameter $w$ that needs to be adjusted, thus increasing the difficulty of tuning.
> Considering these, setting $\mu_Q$ as the average value of the queries is a simple and effective approach.
>
> (iii) For $\beta$, its value is strictly *guaranteed within the range of 0 and 1*, thus preventing numerical overflow.
> As for $\alpha$, it is defined by the formula $\frac{Z}{Z'}$, where $Z$ represents the log-sum-exp of prior logits and $Z'$ represents the log-sum-exp of logits after merging with sparse attention.
> Since sparse attention selects larger logits to replace some of the logits in the prior, $Z'$ is often larger than $Z$.
> Thus, $\alpha$ is *usually well controlled within the range of 0 and 1*, preventing overflow.

---

### Official Review · Reviewer_WqTC · 2025-10-30

**Soundness:** 3
**Presentation:** 3
**Contribution:** 3
**Rating:** 4
**Confidence:** 3

**Summary:**

The paper proposes RESA, a training-free residual estimation framework that complements sparse attention by estimating the contributions of the key–value pairs that sparse attention discards. The key insight is that attention logits are strongly low-rank, allowing a rank-1 prior (computed from the mean query and key vectors) to approximate the global structure of attention. During decoding, RESA merges this prior with sparse attention outputs through an rescaling and delta update, keeping computational cost unchanged. Experiments on LongBench and RULER show that RESA can improve task accuracy and reduce KV budget while maintaining comparable performance.

**Strengths:**

The paper introduces a simple yet effective approach to compensate for the information loss in sparse attention using a low-rank residual estimation. It is training-free, computationally efficient, and integrates seamlessly with existing sparse attention mechanisms. The empirical results are consistent across different models and benchmarks, demonstrating tangible performance and throughput improvements. The low-rank analysis provides interesting insight into the structure of attention logits, which may inspire future theoretical or practical extensions.

**Weaknesses:**

1. The theoretical justification is incomplete. The proposed estimator is intuitive and empirically validated but lacks formal analysis or error bounds. Providing theoretical guarantees or conditions for the rank-1 prior to hold would greatly strengthen the claims.

2. The computation of the prior under chunked prefill remains ambiguous. It is unclear whether the prior is aggregated across chunks or derived from the last chunk only, and the paper lacks a detailed runtime breakdown to justify the claimed negligible overhead.
3. The performance is sensitive to the hyperparameter λ, which controls the influence of the prior. Larger λ values can hurt summarization or comprehension tasks, but there is no guidance on automatic or adaptive tuning strategies.
4. Some tables contain minor formatting errors and inconsistencies. (Table 7 and Table 8, "quert" should be "quest")

**Questions:**

1. How is the prior computed when prefilling is chunked? Is the prior accumulated across all chunks, or only computed at the final chunk? Can you quantify the prefill-time overhead relative to standard sparse attention?

2. Have you explored an adaptive λ schedule, for example, using entropy or sparsity of the attention logits as a signal to modulate prior strength dynamically?

3. Can you compare RESA with low-rank inference baselines such as a learned rank-1 bias or lin-attention variant to clarify its novelty relative to existing low-rank approaches?

4. Summarization and comprehension tasks appear to degrade at larger λ. Can you analyze failure cases and propose mitigation strategies beyond manually reducing λ?

5. For numerical stability, since α and β are computed via log-sum-exp normalization, did you observe pathological heads or tokens with extreme α/β? Were any clipping or smoothing techniques applied?

---

> ### Author Response · Authors · 2025-11-22
> **Response to Reviewer WqTC (Part I)**
>
> We thank the reviewer for raising concerns, we will answer questions unrelated to efficiency below.
> Questions about efficiency can be seen in the **Response to All Reviewers**.
>
> **Q1:**
> More theoretical analyze is needed.
> The author intuitively and empirically discover that the residual attention score has rank-1 nature.
> It would be better if author can provide theoretical analysis for why those residual attention score has those kind of rank-1 nature.
>
> **A1:**
> We provide a clearer and more formal explanation of why our approximation is effective from three aspects:
>
> *1. The prior $\mathbf{\mu_Q K^T}$ represents the mean of historical logits, which aligns closely with the principal SVD component.*
> For query $q_i$, given the true logit $z_i=q_iK^T$,
> our prior $\mu_QK^T=(\frac{1}{L}\sum_{i=1}^Lq_i)K^T=\frac{1}{L}\sum_{i=1}^Lz_i$, which represents *the mean value across the entire prefilling attention logits*.
> Using this as the prior is because that we observed the historical logits of each token (i.e., a column of attention logits matrix) exhibit a concentrated structure: *these logits mostly revolve around their averaged logit with relative small fluctuations (figures will be added in the final version).*
> Thus, the corresponding elements in each row of the attention logits matrix are unlikely to change significantly (if there are significant changes, they will be captured by sparse attention).
> This leads to the *rank-1 spike* in attention logits.
> In such cases, when rows of attention logits matrix share strong similarities at most time, their mean vector is almost collinear with the principal singular vector.
> Thus, $\mu_QK^T$ naturally aligns with the principal SVD component.
> However, note that the mean of $q_i$ here assumes that the attention logits matrix has no causal mask (i.e., a complete square matrix).
> If it has a causal mask (i.e., a lower triangular square matrix), then the more recent the token, the fewer the historical logits, which is more likely to cause fluctuations for prior estimation.
>
> *2. The bias term $(q_i-\mu_Q)\cdot \mu_K^T$ enforces exact alignment of the true logit mean, which avoids the distortions when merging with sparse attention.*
> Given the true logits $q_iK^T$, the mean value of $q_iK^T$ over all keys is $q_i\mu_K^T$, where $\mu_K^T$ is the mean of all keys.
> However, considering our prior estimation $\mu_QK^T$, its mean value is $\mu_Q\mu_K^T$, which is not aligning with the ground truth.
> Thus, our proposed bias term $(q_i-\mu_Q)\cdot \mu_K^T$ provides *an exact correction such that the estimated logits and the true logits share the same mean.*
> *Aligning the mean of prior and ground truth eliminating global bias shifts, which is crucial to avoid systematic distortions of original attention.*
> This is because softmax is very sensitive to relative, not absolute logits, so when aggregating prior and real logits selected by sparse attention, a large global bias will significantly change the distribution of attention scores.
> Thus, the bias term $(q_i-\mu_Q)\cdot \mu_K^T$ can significantly stabilize the approximation, and it is deterministic, training-free and can be adaptively adjusted through online query $q_i$.
>
> *3. The remaining error is a second-order interaction of query/key deviations that is relative small.*
> For $q_i$, the error $\delta$ between true logits and our estimated logits is $\delta=q_iK^T-(\mu_QK^T + (q_i-\mu_Q)\cdot \mu_K^T)=(q_i-\mu_Q)\cdot(K-\mu_K)^T$, which shows that $\delta$ is a bilinear interaction between query deviation and key deviation.
> Hence, it is second-order relative to the dominant mean-based structure.
> In Figure 6(b) of our paper, we visualized this error term and find some mean-reverting patterns.
> Furthermore, in our response to **Reviewer 2Wxj** (A4), we also mentioned a hypothesis that might be related to the fixed bias introduced by RoPE.
> This is an interesting area we are currently exploring.
> (This answer is similar to the response to the **Reviewer 5DL8** A1.)
>
>
> **Q2:**
> The author claimed the overhead is negligible under chunked prefill. Does the prior aggregate across chunks or derived from only the last chunk? Could author explain more about this?
>
> **A2:**
> Prior estimation only occurs in the last chunk and is combined with its computation to make full use of computing power, so it is only calculated once.
> For more detailed evaluation, please refer to the **Response to All Reviewers**.

---

> ### Author Response · Authors · 2025-11-22
> **Response to Reviewer WqTC (Part II)**
>
> **Q3:**
> How to automatically chose the hyperparameter $\lambda$.
> The performance is highly depend on the hyperparameter.
> So is there a automatic way, or recipe for how to choose the hyper-parameter $\lambda$?
>
> **A3:**
> Currently, we primarily tune $\lambda$ manually because we've found that suitable $\lambda$ is often either close to 1 or close to 0).
> Therefore, when tuning, we first try larger (close to 1) $\lambda$; if the effect is unsatisfactory, we adjust to smaller (close to 0) $\lambda$.
> This process does not incur significant time overhead.
> Furthermore, it's worth emphasizing that *the $\lambda$ here are a reasonable abstraction, fully describing the strength of the prior*.
> Meanwhile, *the method of adjusting hyperparameters does not affect the core contributions of RESA, namely, compensating for sparse attention with prior estimation and online aggregation*.
> However, automatically selecting hyperparameters is indeed a direction worthy of further exploration.
>
> To better understand how $\lambda$ affects the accuracy of tasks, we present a case study to show more details and analysis.
>
> **Case Study:**
> Take the CWE task as an example, one of the tasks where accuracy drops most severely with large $\lambda$.
> This task requires the model to identify the top-10 most frequently occurring words in the context.
> When $\lambda$ is large, although the task score decreases, the model does not output nonsense; instead, *it uses more of its internal parameter knowledge than contextual knowledge to answer the question*.
> Specifically, the model does not output high-frequency words from the context, but rather high-frequency English words like "a" and "the", indicating that the model is answering based on its own training corpus (i.e., internal parameter knowledge).
>
> **Analysis:**
> While Prior does interfere with the model's understanding of context (that's why we introduce $\lambda$ to control its strength), further research reveals that *the key problem is not about that the existence of prior is unreasonable, but rather its specific calculation method.*
> In the CWE task, frequency statistics for each word are updated at different positions.
> However, the averaging operation used when calculating $\mu_Q$ for our prior mixes information from different positions, causing confusion for the model, so it can only obtain results based on its own knowledge.
> When we simply choose $\mu_Q$ as the last query in the prefilling stage, CWE's performance  significantly improves from **5.8 to 24.1** (ArkVale) and **5.4 to 21.2** (Quest) with $\lambda=1$.
> *This indicates that the specific calculation method of prior matters and requires further exploration.*
> However, we need to emphasize that RESA's main contribution is a framework that compensates for sparse attention with prior, not selecting the optimal $\mu_Q$.
> This does not affect RESA's core contributions and worth future exploration.
>
>
> **Q4:**
> Some table contains typos. (Table 7 and Table 8, "quert" should be "quest")
>
> **A4:**
> We appreciate the errors pointed out by the reviewer and will make corrections in the revision.

---

### Official Review · Reviewer_2Wxj · 2025-10-31

**Soundness:** 3
**Presentation:** 3
**Contribution:** 3
**Rating:** 6
**Confidence:** 3

**Summary:**

This paper addresses a key limitation of Sparse Attention (SA) methods used in LLM inference. While SA accelerates decoding by selecting and computing only a small subset of "critical" Key-Value (KV) pairs, it completely ignores the contribution of all unselected KVs, leading to a degradation in model quality.

The authors propose RESA (Residual Estimation for Sparse Attention), a novel, training-free framework that "brings back what sparse attention ignores". Instead of just ignoring the unselected KVs, RESA estimates their collective contribution (the "residual") and adds it back to the sparse attention output, thereby improving accuracy.

The core insight is that the attention logits matrix (pre-softmax $QK^T$) is inherently low-rank. The authors show via Singular Value Decomposition (SVD) that the effective rank is upper-bounded by the head dimension (e.g., 128) regardless of sequence length (e.g., 8k), and that the principal singular value (rank-1) is dominant, accounting for 40-50% of the matrix's energy. SA methods excel at capturing fine-grained sparse peaks but discard this global low-rank structure.

**Strengths:**

1. Originality and Significance: The paper's core idea is highly original. The vast majority of SA research focuses on "how to select the top-k KVs more accurately or efficiently". This work introduces a completely orthogonal and complementary perspective: "how to efficiently account for the $N-k$ KVs that were not selected". This reframes the problem from simple sparsity to a more robust sparse + low-rank approximation. The resulting improvements are very significant: boosting accuracy (up to 26%) or saving 33.2% of the KV budget for free is a major practical contribution.

2. RESA's effectiveness is demonstrated across multiple models (Llama-3, Mistral, LWM) and multiple challenging long-context benchmarks (RULER, LongBench).

**Weaknesses:**

Please see Questions below.

**Questions:**

1. Prior Estimator Cost: Could you please quantify the one-time wall-clock latency (in ms) of the "Prior Estimator" step at the end of prefill? How does this cost scale as sequence length L increases from 8k to 128k?

2. Could you please provide a precise definition for the "typical query" $\mu_Q$? Is it computed as the simple average of all $L$ query vectors in the prefill context? How sensitive is the model's accuracy to this choice (e.g., mean of all queries vs. just the last query)?

3. The paper's motivation (low-rank logits) applies to the prefill stage as well (Figure 2a). While the proposed mechanism is for the decoding stage, could a similar "residual estimation" idea be used to accelerate the $O(N^2)$ prefill stage itself?

4. The error of the logit estimation (Fig 6b) shows a clear periodic, mean-reverting pattern. You note this "temporal structure" is worth exploring. Do you have a hypothesis for what causes this periodicity? Does it align with any known properties of RoPE or the model architecture?

---

> ### Author Response · Authors · 2025-11-22
> **Response to Reviewer 2Wxj**
>
> We thank the reviewer for raising concerns, we will answer questions unrelated to efficiency below.
> Questions about efficiency can be seen in the **Response to All Reviewers**.
>
> **Q1:**
> Prior Estimator Cost: Could you please quantify the one-time wall-clock latency (in ms) of the "Prior Estimator" step at the end of prefill? How does this cost scale as sequence length L increases from 8k to 128k?
>
> **A1:**
> For more detailed evaluation, please refer to the **Response to All Reviewers** (the part of prefilling overhead).
>
>
>
> **Q2:**
> Could you please provide a precise definition for the "typical query" $\mu_Q$?
> Is it computed as the simple average of all query vectors in the prefill context?
> How sensitive is the model's accuracy to this choice (e.g., mean of all queries vs. just the last query)?
>
> **A2:**
> In our paper, $\mu_Q$ is indeed defined as the average of all queries.
> First, we need to clarify that such a determination actually utilizes the mean of historical logits as the prior (historical logits often exhibit mean regression characteristics).
> We provided more details and support in our response to **Reviewer WqTC** (A1) or **Reviewer 5DL8** (A1).
> Second, for other methods of choosing $\mu_Q$, such as directly selecting the last query, we experimentally demonstrate that their performance is inferior to the average of the queries. Some results of Llama-3.2-3B is shown below.
>
> | Llama-3.2-3B-8K    | NIAH3    | MK1   | MK2    | MK3    | MV   | MQ |
> |----------------------|-------|-------|--------|--------|--------|--------|
> | ArkVale-mean-q              | 97  | 98 | 79  | 6 | 88 | 94 |
> | ArkVale-last-q             | 89  | 97 | 79  | 2 | 82 | 92.75 |
> | Quest-mean-q              | 84  | 98 | 82  | 5 | 84.25 | 93 |
> | Quest-last-q             | 74 | 94 | 78 | 3 | 74.5 | 89.5 |
>
> This is because the mean of the queries captures global information better.
> However, while there are various other ways to choose $\mu_Q$, we need to emphasize that *RESA's main contribution is as a framework to compensate for the accuracy of sparse attention, not focusing on the specific selection of $\mu_Q$.*
> However, smartly selecting $\mu_Q$ is a direction worth exploring in the future.
>
>
> **Q3:**
> The paper's motivation (low-rank logits) applies to the prefill stage as well (Figure 2a).
> While the proposed mechanism is for the decoding stage, could a similar "residual estimation" idea be used to accelerate the prefill stage itself?
>
> **A3:**
> We believe there is indeed an opportunity to use the prefill stage, but challenges remain.
> During prefilling, it's difficult to accurately determine the appropriate position to start using RESA (e.g., starting at context length 1k, 2k, 4k, etc.), leading to inconsistent model quality.
> Therefore, RESA is actually a more conservative strategy, used only in the decoding stage.
> *Even in the worst case, it guarantees the same effect as the original sparse attention* (i.e., $\lambda=0$), and there are considerable works to ensure its stability.
>
>
> **Q4:**
> The error of the logit estimation (Fig 6b) shows a clear periodic, mean-reverting pattern.
> You note this "temporal structure" is worth exploring.
> Do you have a hypothesis for what causes this periodicity?
> Does it align with any known properties of RoPE or the model architecture?
>
> **A4:**
> While there is no definitive hypothesis, we do believe there is a strong correlation between error and RoPE.
> Specifically, comparing attention logits with and without ROPE, we found that in some heads, *RoPE does provide a near-periodic, relatively fixed bias, which we call the positional bias, and its trend closely resembles that of error*.
> However, this is not universally applicable, and a quantitative relationship cannot be well established.
> However, we are now conducting further investigations to reveal the deeper relationship.

---

### Author Response · Authors · 2025-11-22
**Response to All Reviewers about Extra Overheads**

We thank the reviewers for raising concerns about efficiency of RESA compared to standard sparse attention.
Below we provide a unified clarification from three aspects: prefilling overhead, decoding overhead and memory overhead.

**Prefilling Overhead is Small.**
The extra overhead of prefilling mainly comes from the prior estimation process of RESA.
First, we need to clarify that in practice, the *calculation of prior is merged into the prefilling to fully utilize computing power*, rather than being performed separately.
Thus, given a context length of $L$, the actual calculated context length is $L+1$.
Second, we used the chunked prefill technique to avoid memory explosion.
Therefore, the calculation of *prior is merged with the calculation of the last chunk*, which indicates that using RESA will increase the chunk size by 1.
We measure the extra overheads in two ways:
1. comparing the time of the last chunk w/ and w/o RESA.
2. Comparing the time of the entire prefilling w/ and w/o RESA.

For the first one, given chunk size equals to 1024 in our paper, the layer-wise latency (ms) of the last chunk with context length from 8k to 128k is shown below, with each test runs 100 times and averaged.

| context length   | 8k    | 16k   | 32k   | 64k   | 128k  |
|---------------------|-------|-------|-------|-------|-------|
| w/o RESA             | 6.8   | 13.3  | 25.0    | 48.7  | 96.0    |
| w/ RESA            | 7.3   | 14.0    | 26.2  | 51.1  | 100.6 |
| extra overhead rate | 7.35% | 5.26% | 4.80% | 4.93% | 4.79% |

It can be seen that the ratio of extra overhead gradually decreases and tends to stabilize.
For the second aspect, the overall latency (ms) is shown below, with each test run 100 times and averaged.

| context length    | 8k    | 16k   | 32k    | 64k    | 128k   |
|----------------------|-------|-------|--------|--------|--------|
| w/o RESA              | 27.3  | 106.3 | 399.8  | 1554.8 | 6140.4 |
| w/ RESA             | 27.8  | 106.9 | 400.8  | 1556.7 | 6143.8 |
| extra overhead rate  | 1.83% | 0.56% | 0.25%  | 0.12%  | 0.06%  |

It can be seen that the overheads are almost negligible.
This is because the longer the context, *the more chunks need to be computed, while RESA only applies prior estimation to the last chunk.*

**Decoding Overhead is Small.**
The extra overhead of decoding mainly comes from the Online Aggregator, which merges the prior and real logits selected by sparse attention.
Although our paper demonstrates that its computational complexity is consistent with sparse attention, there may still be a constant-fold difference.
Therefore, given the same KV budget, we compare the layer-wise decoding latency (ms) of each step w/ and w/o RESA (the same KV budget 500, using Quest), and the results are shown below, with each test run 100 times and averaged.

| context length    | 8k    | 16k   | 24k    | 32k    |
|----------------------|-------|-------|--------|--------|
| w/o RESA              | 1.92  | 2.81 | 3.68  | 4.28 |
| w/ RESA             | 1.98  | 2.89 | 3.77  | 4.38 |
| extra overhead rate  | 3.13% | 2.85% | 2.45%  | 2.34%  |

It can be seen that the extra overhead is less than 3.13\% for each step.
As the sequence increases, the additional overhead gradually decreases.
This is because *all the extra computations of the Online Aggregator are lightweight, element-wise multiplication or addition and do not involve any matrix multiplication*, which can be clearly seen from the Equation 5 in our paper.
Note that we have not yet implemented extreme system optimizations (e.g., operator fusion), so this overhead can be reduced further or even achieve zero cost.
However, despite the very slight overhead at each step of decoding, using RESA improves the overall efficiency.
This is because, while maintaining the same accuracy, RESA reduces the required KV budget, resulting in the benefit of reduced memory I/O overhead outweighing the extra overhead of decoding.
Figure 7(b) in our paper discusses the benefits of RESA by comparing PSA and Ada-KV.
We will provide more detailed breakdowns in the final version.

**Memory Overhead is Small.**
For a specific layer, the extra memory overhead mainly consists of four parts:
1. The prior logits with shape $(B, H, L, 1)$.
2. The output $O_{est}$ corresponds to the prior with shape $(B, H, 1, d)$.
3. The mean of queries $\mu_Q$ with shape $(B, H, 1, d)$.
4. The mean of keys $\mu_K$ with shape $(B, H, 1, d)$.

$B, H, L, d$ denote batch size, number of head, sequence length and head dim.
Considering that the original KV cache size is $(B, H, L, d)$, therefore, the ratio of extra memory overhead is: $r = \frac{1}{d} + \frac{3}{L}$.
When $L$ is very large, $r\approx\frac{1}{d}$.
Since a common configuration of $d$ is 128, 256 etc., the *extra memory overhead is less than 1\% and can be ignored.*

---

### Author Response · Authors · 2025-12-03
**Our Key Contributions and Summary of Major Revisions**

We sincerely thank all the reviewers for their valuable comments.


## Our key contributions:

1. We clearly identify the **redundancy of attention logits matrix** (before softmax), providing **a new low-rank perspective beyond mainstream sparse attention** methods to improve performance.

2. We propose RESA, **a training-free framework capturing the approximated low-rank structure during prefilling to compensate all subsequent decoding steps**. RESA achieves this by two modules: a Prior Estimator estimates a prior distribution as the low-rank structure, and an Online Aggregator merges the prior with sparse attention. Both modules are **efficient and totally training-free**.

3. We further analyze the effectiveness of RESA comes from **the prior being used as the bias of attention**, which can be used to inject knowledge and is worth future exploration at training-time.

4. We conduct experiments to show the effectiveness of RESA, including the **accuracy improvement** on two famous benchmarks and the **KV budget reduction**. We also conduct extensive experiments, including **error analysis, hyperparameter tuning, and extra overhead analysis**, to better demonstrate the advantages of RESA.


## Revision:

The reviewers' comments primarily focused on two aspects:

1. **Insufficient analysis of efficiency**, including the extra overhead of the prefilling stage, the extra overhead of the decoding stage, and the extra overhead of memory.

2. **Insufficient theoretical analysis of the prior estimation**, requiring more supporting evidence and analysis.

Therefore, our revision mainly focuses on the above two aspects, including rewriting and providing more detailed material and content. Specifically, we made the following changes in the revision (all changes are **marked in blue**):

1. **Adding a more theoretical and formulated support** to our rank-1 approximation estimation. (Section 4.2)

2.  **Adding more detailed experiments and analysis about the extra overhead** of prefilling, decoding and memory, proving RESA is a lightweight, training-free method to enhance sparse attention. (Section 5.5)

3. Fix some typos mentioned by Reviewers (Appendix Table 9 to Table 12).

---

### Meta-Review · Area_Chair_qVTJ · 2026-01-06

**Summary:**

### Strength emphasized by reviewers
1. This paper takes a novel pathway to improve the sparse attention methods, that is compensating the information lost by sparse attention.
2. The empirical observations to support the method design are clear and solid.
3. This method is training-free, and thus easier to apply.
4. Effectiveness is demonstrated across multiple model families and benchmarks.
5. The insights found by this paper are interesting and might inspire future work.

### Ensemble of concerns and suggestions
1. Insufficient analysis of efficiency [Reviewer 2Wxj,WqTC,WmEf,5DL8].
2. Insufficient theoretical analysis [Reviewer WqTC,5DL8]
3. The performance seems to be sensitive to the hyperparameter $\lambda$ [Reviewer WqTC,WmEf].
4. Suggest the authors to compare RESA with low-rank inference baselines such as learned rank-1 bias or lin-attention varaint [Reviewer WqTC].
5. Lack evaluation on more complex tasks besides retrieval and simple QA [Reviewer WmEf].

### My additional comments and questions for the authors
Overall, this paper is novel, solid, and provide some insights for future study. But there are still concerns in the paper writing as well as the method that the authors should address or at least discuss in the final revision:
1. In section 5.4, the results are shown in Figure 7(a) instead of Figure 15(a). The authors should fix this in the final revision.
2. How does RESA apply to long-context scenarios where the input prompt is short but the output is very long? In this case, estimating an $\mu_Q$ and $\mu_K$ after the prefilling stage might not be so useful in providing meaningful information? Therefore, at which position should the typical query be computed?
3. I suggest the authors to explicitly write the error $\delta$'s formula (as you did in the response to Reviewer 5DL8), and also explicitly write the formula for computing $P$ (i.e., explicitly mentioning the right part of Eq.2 in the paper). This will make the paper read more smoothly.
4. I suggest the authors to finish adding experiment on more benchmarks as they said in the response to Reviewer WmEf: "trying more experiments and will add the necessary experiments in the final version".
5. I suggest the authors to include the rebuttal discussion (with Reviewer WqTC and WmEf) on the adjustment of the $\lambda$ and other options of calculating $\mu_Q$ in the final revision.

**Reviewer Concerns:**

The author rebuttal addressed Concern 1, 2, 3, but not addressed Suggestion 4, 5: The authors didn't respond to Suggestion 4 (about more baselines). For Suggestion 5, the author response said "trying more experiments and will add the necessary experiments in the final version".

**Reviewer Scores:**

I think the reviewers will at least not decrease the score, for Reviewer WqTC (the only reviewer that gives a negative score 4), I think the rebuttal revision and author response successfully answered most of the reviewer concerns and questions, except that the authors do not compare with additional baselines (e.g., learned rank-1 bias). There is some chance Reviewer WqTC will increase the score.

---

> ### Public Comment · ~Weihao_Yang1 · 2026-03-02
> **Response To Area Chair qVTJ**
>
> We sincerely thank you for your valuable comments. Here we make a brief summary of our camera ready version.
>
> For Q1, we fixed the typo in our final revision.
>
> For Q2, actually it is the commonly seen reasoning/thinking model nowadays, which is often studied differently from traditional models. Although the idea that considering low-rank as compensation revealed by RESA is still valid, in our experiments calculating typical query only at the end of prefilling does not show significant benefits. Therefore, this can indeed be considered as a direction for our future improvements.
>
> For Q3, we added the formula of error $\delta$ and P in our paper.
>
> For Q4, we added more evaluations about RESA in our appendix (QA and coding), while reasoning needs further exploration as mentioned above. Besides, reviewer WqTC also mentioned about learned rank-1 bias or linear attention, which often requires retraining or training from scratch. However, RESA is a totally training free method that not suitable for comparison.
>
> For Q5, we add the rebuttal discussion in our appendix due to the page limit.

---

### Decision · Program_Chairs · 2026-01-26

Accept (Poster)